# Constrained Adaptive Rejection Sampling

Paweł Parys [1] [*]   Sairam Vaidya [2] [*]   Taylor Berg-Kirkpatrick [2]   Loris D'Antoni [2]

## Abstract

Language Models (LMs) are increasingly used in applications where generated outputs must satisfy strict semantic or syntactic constraints. Existing approaches to constrained generation fall along a spectrum: greedy constrained decoding methods enforce validity during decoding but distort the LM's distribution, while rejection sampling (RS) preserves fidelity but wastes computation by discarding invalid outputs. Both extremes are problematic in domains such as program fuzzing, where both *validity* and *diversity* of samples are essential. We present *Constrained Adaptive Rejection Sampling* (CARS), an approach that strictly improves the sample-efficiency of RS without distributional distortion. CARS begins with unconstrained LM sampling and adaptively rules out constraint-violating continuations by recording them in a trie and subtracting their probability mass from future draws. This adaptive pruning ensures that prefixes proven invalid are never revisited, acceptance rates improve monotonically, and the resulting samples exactly follow the constrained distribution. In experiments on a variety of domains—e.g., program fuzzing and molecular generation—CARS consistently achieves higher efficiency—measured in the number of LM forward passes per valid sample—while also producing stronger sample diversity than both Greedy Constrained Decoding (GCD) and methods that approximate the LM's distribution.

## 1. Introduction

Many applications of Language Models (LMs) require outputs that are not just fluent, but also satisfy strict structural or semantic constraints (Geng et al., 2025). Examples include ensuring syntactic validity in programming languages, adherence to schemas in data formats, or generating programs in restricted fragments of a given language.

This issue has motivated extensive work on *constrained generation*, i.e., methods for sampling using a language model so that its outputs satisfy a given structural or semantic specification. Two fundamental requirements emerge in this problem space: *(i)* **Fidelity:** do samples follow the *exact* LM distribution conditioned on the constraint, or only an approximation? *(ii)* **Efficiency:** how many LM forward passes are required to obtain valid samples?

Most existing methods, which fall into three families, succeed on one axis but sacrifice the other.

**Exact methods.** Rejection Sampling (RS) is the canonical example. It produces unbiased samples from the true constrained distribution but wastes computation by discarding the overwhelming majority of candidates (e.g., $< 1\%$ acceptance in many structured domains).

**Static approximation methods.** Greedy constrained decoding (GCD) enforces validity by masking tokens that lead to constraint failure during generation (Geng et al., 2023; Park et al., 2025). While efficient, GCD provably distorts the conditional distribution (Tam et al., 2024; Park et al., 2024), often degrading downstream performance (Tam et al., 2024). It can even fail to terminate in some cases (e.g., repeatedly "opening brackets" without producing a complete valid sequence).

**Asymptotic approximation methods.** These methods include iterative over-approximations of invalid prefixes (Park et al., 2024), Monte Carlo and sequential Monte Carlo approaches that resample from inexact constrained distributions to approximate the desired distribution (Anaya Gonzalez et al., 2025), and other MCMC-style refinements (Lew et al., 2023; Melcer et al., 2024b). All of these techniques are guaranteed to converge to the correct distribution in the limit, but they provide no principled stopping rule: early samples can be arbitrarily biased, and efficiency depends heavily on how many candidates must be drawn before the approximation stabilizes. Moreover, these methods require hyperparameter tuning (e.g., number of MCMC steps $k$ or SMC particles $M$) without providing practitioners with a principled way to determine if these hyperparameters yield

---
[*]Equal contribution   [1]University of Warsaw, Poland   [2]Department of Computer Science and Engineering, University of California-San Diego, San Diego, USA. Correspondence to: Sairam Vaidya <smahadevaganapathy@ucsd.edu>.

*Proceedings of the 43rd International Conference on Machine Learning*, Seoul, South Korea. PMLR 306, 2026. Copyright 2026 by the author(s).

sufficient distribution fidelity.

The current landscape reflects a fundamental tradeoff: *exactness without efficiency, or efficiency without exactness*. This tradeoff becomes especially limiting when performance depends on generating sets of diverse, constraint-satisfying outputs from the same LM context—such as program fuzzing (Anaya Gonzalez et al., 2025) or molecule discovery (Wang et al., 2023). In these cases, the key desideratum is not only fidelity, but also **amortized efficiency** across many samples. Existing asymptotic methods achieve amortized efficiency only asymptotically, and only at the cost of biased early samples. *What is missing, is an exact algorithm that is amortized-efficient in practice.*

We propose *Constrained Adaptive Rejection Sampling* (CARS), an exact method that combines the fidelity of RS with the efficiency benefits of constraint-aware decoding. CARS builds on Adaptive Rejection Sampling (ARS) (Gilks & Wild, 2018; Mansinghka et al., 2009), which avoids repeating rejected samples. CARS goes further: as each sample is generated, CARS uses constrained decoding algorithms to identify not only the rejected output but also all nearby continuations of its partial prefixes that would inevitably violate the constraint. Each invalid prefix is recorded in a trie, and its probability mass is subtracted from future generations, ensuring monotonic improvements in acceptance rate while preserving the exact constrained distribution.

Although, in theory, CARS could still require many rejections for adversarial constraints, we argue and demonstrate empirically that real-world constrained LM tasks fit the CARS setting well: most constraints are prefix-checkable (e.g., validity according to a context-free grammar or type system) and highly informative for pruning. This makes CARS asymptotically efficient in practice while remaining exact, thus setting a new state-of-the-art for sampling from the exact LM's constrained distribution.

We make the following contributions. We introduce **CARS**, a new algorithm for constrained LM generation that achieves exactness with practical efficiency by leveraging constraint structure (Section 3). We show that CARS achieves higher acceptance rates, stronger diversity, and lower amortized cost than existing constrained sampling methods (Section 4).

## 2. Exact Constrained Sampling

In this section, we formalize the problem of sampling from a language model (LM) conditioned on a constraint (i.e., constrained sampling), define our key desiderata of a good constrained sampling algorithm, and describe how existing constrained sampling algorithms do not meet such desiderata. We follow the definitions proposed by Park et al. (2024) and Anaya Gonzalez et al. (2025).

Let $\Sigma_\$$ be a set of tokens including an end-of-sequence marker $, and let $\Sigma = \Sigma_\$ \setminus \{\$\}$. We consider sequences from the set $\Sigma^*\$^?$ (i.e., sequences of tokens that may have $ only at the end). We write $u \preceq w$ to denote that a sequence $u$ is a prefix of a sequence $w$. For a set of sequences $\mathcal{L}$ we write $\mathrm{prefix}(\mathcal{L})$ to denote the set of prefixes of sequences in $\mathcal{L}$—i.e., $\mathrm{prefix}(\mathcal{L}) = \{u \mid \exists w \in \mathcal{L}.\, u \preceq w\}$—and $\mathrm{ext}(\mathcal{L})$ to denote sequences extending a sequence from $\mathcal{L}$—i.e., $\mathrm{ext}(\mathcal{L}) = \{w \in \Sigma^*\$^? \mid \exists u \in \mathcal{L}.\, u \preceq w\}$.

**Language Models.** An (autoregressive) language model is given by next-token conditional probability distributions of the form $P(ua \mid u)$, where $u \in \Sigma^*$ and $a \in \Sigma_\$$—denoting the probability that a sequence $u$ is followed by a token $a$. This definition extends to longer continuations: $P(ua_1 \ldots a_n \mid u) = \Pi_{i=1}^n P(ua_1 \ldots a_i \mid ua_1 \ldots a_{i-1})$.

More generally, for any prefix $u \in \Sigma^*$ and suffix $w \in \Sigma^*\$^?$, we write $P(w \mid u)$ for the probability that the model generates $w$ as a continuation of $u$ before either producing the end-of-sequence symbol $ or reaching length $|w|$. We also write $P(w \mid u) = 0$ when $u$ is not a prefix of $w$. For technical reasons, we assume that $\sum_{w \in \Sigma^*\$} P(w) = 1$, which means that almost surely the $ marker will be produced at some moment (the probability that an infinite word without any $ marker will be produced is $0$; this can be achieved by modifying the LM to generate $ with probability $1$ after a maximum sequence length).

**Constraints.** Given a language model $P$ and a constraint, the goal of constrained sampling is to sample sequences that satisfy the constraint. Formally, a constraint is just a set $\mathcal{L} \subseteq \Sigma^*\$$ of sequences (satisfying the constraint). In practice, the set $\mathcal{L}$ may be given in many possible ways, e.g., as a regular language, context-free grammar (CFG), or some logical condition.

While some constraints are computationally expensive to verify, in this work and in our experiments, we focus on constraints that can be incrementally evaluated over the entire token vocabulary. This means that we have a fast algorithm that given a word prefix $u$ generates a vector of answers, saying for each possible next token $a \in \Sigma_\$$ whether $ua \in \mathrm{prefix}(\mathcal{L})$ (i.e., whether $ua$ can be continued into a full sequence satisfying the constraint). In particular, this holds for context-free grammars (CFGs) (AI, 2025; Park et al., 2025), which can, for instance, describe the set of syntactically valid programs in a programming language or enforce the correct structure of a JSON object.

An example domain where constrained decoding is used to generate many diverse samples is asking a language model to generate SQLite regression test files that exercise as many distinct execution paths in the SQLite engine as possible (see Section 4.1). To target specific components of the

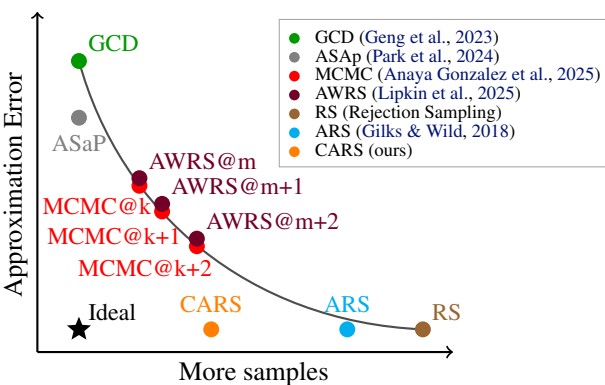

*Figure 1.* Comparison of CARS to other exact and inexact methods.

---

**Algorithm 1** CARS algorithm

1: **Input:** Constraint language $\mathcal{L} \subseteq \Sigma^*$
2: **Output:** Infinite sequence of samples drawn from the constrained distribution $P^{\mathcal{L}}$
3: $\mathcal{W} \leftarrow \emptyset$      {*initialize invalid prefixes*}
4: **while** true **do**
5:     {$R^{\mathcal{W}}$ is reshaped to avoid invalid samples in $\mathcal{W}$}
6:     $w \sim R^{\mathcal{W}}$     {*sample from reweighted distribution*}
7:     **if** $w \in \mathcal{L}$ **then**
8:        **yield** $w$     {*yield a sample*}
9:     **end if**
10:    $\mathcal{W} \leftarrow \mathcal{W} \cup \text{INVALID}(w, \mathcal{L})$    {*add invalid prefixes*}
11: **end while**

---

database, each file must satisfy the syntactic and semantic rules of the SQLite test-script grammar.

**Exact Constraint-Aligned Sampling.** Constraint-aligned sampling aims to generate sequences from a model $P$ that satisfy a given set of hard constraints, while preserving the model's underlying distribution. Formally, this corresponds to sampling sequences from the constrain set $\mathcal{L}$, where the probability of each word $w \in \mathcal{L}$ should be $P^{\mathcal{L}}(w) = \frac{P(w)}{\sum_{w' \in \mathcal{L}} P(w')}$.

In this work, we focus on designing an algorithm that samples **exactly from the conditional distribution $P^{\mathcal{L}}$** while being **more efficient than existing exact methods**.

**Existing Exact Methods.** Rejection Sampling (RS) repeatedly draws outputs from the LM and discards those violating the constraint. RS is inefficient when valid sequences are rare under the LM, e.g., in structured domains. Adaptive Rejection Sampling (ARS) (Gilks & Wild, 2018; Mansinghka et al., 2009) improves upon RS by dynamically avoiding previously observed invalid samples. It remains exact but only adapts to prefixes or outputs that have been explicitly seen to fail.

Exploiting additional prefixes is the key distinguishing factor that makes Constrained Adaptive Rejection Sampling (CARS) more efficient. For instance, in our fuzzing benchmarks we observe cases in which ARS maintains a rejection rate higher than 99% even after 1,000 samples, whereas CARS lowers rejection to rates in the 70-95% range after just 100 samples (Section 4.1). Figure 1 clarifies that among all the exact methods, CARS is the most efficient.

## 3. Constrained Adaptive Rejection Sampling

The Constrained Adaptive Rejection Sampling (CARS) algorithm maintains a set $\mathcal{W}$ to rule out invalid prefixes that have been discovered during sampling, and uses the proba-

bility of such prefixes according to the LM to compute an adaptively reshaped version $R^{\mathcal{W}}$ of the sampling distribution that is such that future sampling iterations will provably not repeat past mistakes. This lets us retain the exact distributional fidelity of rejection sampling while avoiding wasted computation on already-eliminated sequences .

*Example* 3.1 (Arithmetic Expressions)*.* Our running example uses a grammar for arithmetic expressions over digits:

$$E \ ::= \ d\$ \ | \ d + E \quad \text{where } d \in \{0, 1\}.$$

Here, strings like `1+0+1` satisfy the constraint—i.e., they are accepted by the grammar—while `0++` or `+1` do not.

The rest of the section explains Algorithm 1: how $R^{\mathcal{W}}$ is defined, how the prefix set $\mathcal{W}$ is maintained, and how different update strategies for $\mathcal{W}$ provide different benefits.

**Tracking Invalid Prefixes.** CARS maintains a finite set $\mathcal{W} \subseteq \Sigma^*\$^?$, called *invalid prefixes*. By construction, $\mathcal{W}$ is disjoint from $\text{prefix}(\mathcal{L})$, the set of valid prefixes. For each sequence $u \in \Sigma^*\$^?$, the algorithm implicitly tracks a value $p_u$ representing the probability of extending $u$ into a complete sequence that avoids $\mathcal{W}$:

$$p_u = \sum_{w \in \Sigma^*\$\backslash\text{ext}(\mathcal{W})} P(w \mid u).$$

The values $p_u$ are updated whenever $\mathcal{W}$ is updated. These values satisfy the following equation for words without the end-of-sequence marker $\$$:

$$\forall u \in \Sigma^* \qquad p_u = \sum_{a \in \Sigma_\$} P(ua \mid u) \cdot p_{ua}. \qquad (1)$$

We additionally observe that,

$$p_u = 0 \quad \text{if } u \text{ starts with a known invalid prefix,} \qquad (2)$$
$$\text{i.e., } u \in \text{ext}(\mathcal{W}).$$

$$p_u = 1 \quad \text{if } u \text{ cannot be extended to any known} \qquad (3)$$
$$\text{invalid prefix, i.e., } u \notin \text{prefix}(\mathcal{W}).$$

In the grammar from Example 3.1, we may at some point discover that `0++` is invalid. Then Line 10 adds this prefix to $\mathcal{W}$, and thus any string $u$ that has prefix `0++` has $p_u = 0$.

Initially, we have $\mathcal{W} = \emptyset$, and hence $p_u = 1$ for all sequences $u$—i.e., we have not yet proven any sequence can violate the constraint and their probability of extending to a constraint-satisfying sequence is still upper-bounded by 1.

**The Distribution $R^{\mathcal{W}}$.** At any iteration, given the current set $\mathcal{W}$, CARS samples from a reweighted distribution $R^{\mathcal{W}}$ over the set of sequences $\Sigma^* \setminus \text{ext}(\mathcal{W})$ has not been proven incorrect so far. It is convenient to represent the probabilities associated to each prefix in $\text{prefix}(\mathcal{W})$ using a trie structure. Elements of $\mathcal{W}$ are leaves of the trie, and for internal nodes we store the actual values of $p_u$ calculated according to Equation (1). Note that we do not need to store any more values of $p_u$, as for other sequences we have that either $p_u = 0$ or $p_u = 1$.

When a new sequence $w$ is added to $\mathcal{W}$, we add the corresponding leaf to the trie and set its probability $p_w$ to 0. This update is then propagated upward in the trie: whenever a child probability $p_{ua}$ decreases by $x$, the parent $p_u$ decreases by $P(ua \mid u) \cdot x$. The complexity of an update is linear in the length of $w$ (see Appendix F). We provide an empirical analysis of the trie memory usage in Appendix F.6, where we observe that, in practice, trie growth is sublinear in the number of samples.

For example, suppose `0++` is added to $\mathcal{W}$. The trie node corresponding to `0++` becomes a leaf ($p_{0++} = 0$). Then $p_{0+}$ is decreased proportionally to the probability of extending `0+` with another `+`, thus subtracting the probability of entering this invalid path (which the trie will now disallow).

For a given set $\mathcal{W}$, the quantity

$$p_\varepsilon = \sum_{w' \in \Sigma^* \$ \setminus \text{ext}(\mathcal{W})} P(w')$$

determines the total probability of all sequences avoiding $\mathcal{W}$; we can then define the distribution $R^{\mathcal{W}}$ on sequences $w \in \Sigma^* \$ \setminus \text{ext}(\mathcal{W})$ to be $R^{\mathcal{W}}(w) = \frac{P(w)}{p_\varepsilon}$. The probabilities sum to 1. Importantly, we can sample from $R^{\mathcal{W}}$ left-to-right: for $u \in \Sigma^*$ and $a \in \Sigma_\$$, $R^{\mathcal{W}}(ua \mid u) = P(ua \mid u) \cdot \frac{p_{ua}}{p_u}$.

In our arithmetic-expression grammar, once `0++` is ruled out, whenever the prefix `0+` is visited, the probability of sampling another `+` vanishes, and the model is effectively forced to choose some token other than `+` instead.

Because $\mathcal{W}$ is finite, so is $\text{prefix}(\mathcal{W})$. When we sample a sequence prefix $u$ that does not belong to $\text{prefix}(\mathcal{W})$, we have $p_{ua} = p_u = 1$ (and so on for any extension of $ua$) and $R^{\mathcal{W}}$ reduces to the original distribution $P$. Thus, sampling from $R^{\mathcal{W}}$ almost surely terminates with the $ token.

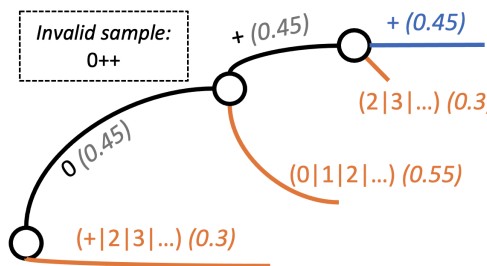

*Figure 2.* Invalid sample `0++` for the arithmetic grammar in Example 3.1. The sequence ending in the blue token is invalid for both ARS and CARS, whereas the sequences ending with orange tokens are only considered invalid by CARS. With the example probabilities in parenthesis, ARS reduces the future rejection probability by $0.09 \approx 0.45 * 0.45 * 0.45$ whereas CARS reduces it by $0.63 \approx 0.3 + 0.45 * 0.55 + 0.45 * 0.45 * 0.45$.

**Updating $\mathcal{W}$.** The update $\mathcal{W} \leftarrow \mathcal{W} \cup \text{INVALID}(w, \mathcal{L})$ at Line 10 determines the efficiency of CARS. Any strategy for updating $\mathcal{W}$ is valid provided that only prefixes outside of $\text{prefix}(\mathcal{L})$ are added to $\mathcal{W}$. Existing rejection sampling approaches can be framed as update strategies:

*Rejection Sampling (RS):* never updates $\mathcal{W}$.

*Adaptive Rejection Sampling (ARS):* adds only the rejected string $w$ or its shortest invalid prefix to $\mathcal{W}$. In the arithmetic-expression grammar, if the sampler produces `0+++`, ARS only adds the prefix `0++` to $\mathcal{W}$ (Figure 2).

*Rejection Sampling with constrained First Token (RSFT):* a variant of ARS that limits invalid prefixes to length 1. When a sequence is rejected, RSFT only adds single-token invalid prefixes to $\mathcal{W}$, ensuring future samples never start with an invalid token while allowing subsequent tokens to be sampled freely. We adopt this method as a baseline in our evaluation to assess how much of the probability mass is "wasted" on sequences with invalid starting tokens. In the arithmetic-expression grammar, regardless of the produced sample, RSFT adds the prefixes `+`, `2`, etc. to $\mathcal{W}$, preventing any sequence from *starting* with an invalid token.

*Constrained Adaptive Rejection Sampling (CARS):* the update strategy that *(i)* adds to $\mathcal{W}$ the shortest prefix $u$ of $w$ that is not in $\text{prefix}(\mathcal{L})$, and *(ii)* for every proper prefix $u$ of $w$ and for every token $a$ such that $ua \notin \text{prefix}(\mathcal{L})$, adds $ua$ to $\mathcal{W}$. In the arithmetic-expression grammar, if the LM produces `0++`, then CARS adds the shortest invalid prefix `0++` to $\mathcal{W}$, but also all invalid continuations of its shorter prefixes—e.g., `+`, `2`, `3`, . . . (invalid continuations of the empty prefix), `01`, . . . (invalid continuations of the prefix `0`), and `0+a`, `0++`, . . . (invalid continuations of the prefix `0+`). Point (ii) applies even if the LM produces a valid sample, e.g., while producing a valid sequence `0+1$`, the same prefixes and their invalid continuations are added to $\mathcal{W}$.

Remarkably, CARS does not require a perfect prefix-validity oracle. At positions where prefix checking is unavailable or too expensive, CARS can simply skip the trie update for that position—i.e., there is a spectrum of update strategies between full-blown CARS and ARS. Exactness is preserved regardless.

**CARS is exact.** Proofs of the following theorems are provided in Appendix E. First, CARS produces unbiased samples from the true constrained distribution and the sample-acceptance rate increases monotonically.

**Theorem 3.2** (Soundness and Monotonicity). *The CARS algorithm samples an element of $\mathcal{L}$ according to the target distribution $P^{\mathcal{L}}$. Moreover, the adaptive updates performed in Line 10 of the algorithm monotonically increase the probability that some sequence is yielded in Line 8 at subsequent loop iterations.*

Although CARS is provably better than ARS, the exact gain is hard to analyze analytically as it depends on both the grammar's branching structure and the LM's token distribution over invalid regions.

The following theorem shows that the expected probability mass removed per sample is non-increasing (i.e., earlier samples are more useful than later ones), which suggests a practical heuristic for running CARS in practice: if subsequent iterations have removed little rejection probability mass, the trie can be frozen and sampling can continue from the fixed distribution $R^{\mathcal{W}}$ without sacrificing significant further improvements.

**Theorem 3.3** (Diminishing Returns). *Let $X_k$ be the random variable saying how much probability mass is removed when the algorithm samples a sequence for the $k$-th time (i.e., it is the value subtracted from $p_\varepsilon$ in that step). Then $\mathbb{E}X_k$ is non-increasing when $k$ grows.*

## 4. Evaluation

In this section, we evaluate CARS in terms of efficiency and the quality of its samples compared to other constrained sampling methods. Because CARS samples exactly from the target grammar-constrained distribution $P^{\mathcal{L}}$, there is no convergence issue. Instead, our focus is on: (i) how efficiently each method produces valid sequences, and (ii) how closely approximate methods (e.g., GCD) match the exact distribution produced by CARS. We evaluate on tasks that require generating many diverse outputs, as this setting best showcases and evaluates amortized efficiency.

In Section 4.1, we demonstrate that seeds generated using CARS improve coverage in fuzzing tasks over approximate methods. Section 4.2 extends the evaluation to molecular synthesis, again highlighting efficiency and constraint satisfaction in domains where diversity is crucial. Section 4.3

evaluates text-to-SQL generation, demonstrating that distributional fidelity translates to improved downstream execution accuracy while maintaining sample efficiency.

Appendix J and K provide additional evaluations on task-focused domains (PDDL planning, SyGuS benchmarks).

**Models.** We evaluate on four models—Llama-3.1-8B-Instruct (Grattafiori et al., 2024), Qwen2.5-7B-Instruct (Qwen et al., 2025), Qwen2.5-14B-Instruct (Qwen et al., 2025), and Llama-3.1-70B-Instruct (Grattafiori et al., 2024). We provide some results in the main body and complete details in Appendix D. Experiments on the 7–14B models are run on a single NVIDIA RTX A6000, while the 70B experiments use 2×NVIDIA H100 GPUs; all results use an optimized inference backend (details in Appendix B). Results for Llama-3.1-70B-Instruct on SMILES and text-to-SQL are deferred to Appendix M.

**Baselines.** We compare CARS against GCD (which is a static inexact approximation), existing exact algorithms discussed in Section 3 (Rejection Sampling (RS), Adaptive Rejection Sampling (ARS) (Gilks & Wild, 2018; Mansinghka et al., 2009), Rejection Sampling with constrained First Token (RSFT)), and MCMC (Anaya Gonzalez et al., 2025), a state-of-the-art approximate algorithm. Among the approximate methods, we do not consider Adaptive Sampling with Approximate expected futures (ASAp) (Park et al., 2024) as Anaya Gonzalez et al. (2025) have shown ASAp always provides worse estimates of the conditional distribution than those given via MCMC. For the benchmarks in Section 4.2, Section 4.3 and Appendix J, we additionally evaluate Adaptive Weighted Rejection Sampling with Sequential Monte Carlo (AWRS) (Lipkin et al., 2025). These benchmarks were considered in the AWRS paper, and the implementation of AWRS directly works on them. The RSFT algorithm (which only learns how to avoid incorrect first tokens) is a special case of CARS that is also a contribution of our work. We select the best settings from the original papers: $k = 10$ steps for MCMC and $M = 10$ particles for AWRS.

**Metrics Shared Across Benchmarks.** Our key metric is sampling efficiency: the number of LM generation calls needed to obtain a fixed number of valid outputs. This metric captures the computational cost of each method and highlights how strategies such as CARS reduce wasted computation on invalid sequences.

To evaluate the distribution fidelity of approximate sampling approaches, the ideal metric would be the distance between the empirical sample distribution and the target constrained distribution $P^{\mathcal{L}}$. However, computing this quantity exactly is impractical: the sequence space is often infinite, and $P^{\mathcal{L}}$ may be inaccessible for direct probability evaluation. We follow the approach by prior work (Park et al., 2024; Anaya

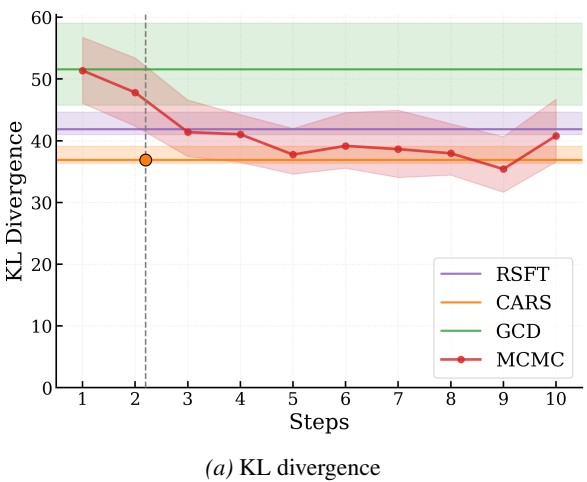

*(a)* KL divergence

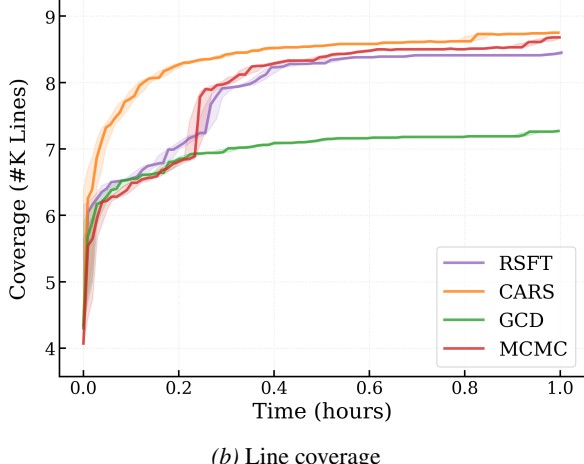

*(b)* Line coverage

*Figure 3.* XML benchmark: *(a)* KL divergence. The KL for exact methods (RSFT and CARS) is non-zero because it is an empirical estimate. The vertical dashed line is the average number of steps MCMC would require to have the same sample efficiency as CARS (i.e., CARS averages 2.15 LM calls per sample) *(b)* Line coverage when fuzzing with generated seeds.

Gonzalez et al., 2025) and use an approximate measure: the Kullback-Leibler (KL) divergence (Kullback & Leibler, 1951) between the empirical distribution of the generated samples $\tilde{P}^{\mathcal{L}}$ and the LM's distribution $P$. We obtain 100 samples for each sampling method for each task, and plot the mean KL divergence and 95% confidence interval ranges computed from bootstrapping across three different runs. Note that the empirical KL divergence can be greater than 0 even when we sample from the exact distribution.

CARS's trie operations account for only a small fraction of total execution time so we defer the analysis of this overhead to Appendix F.

### 4.1. Grammar-based Fuzzing

Anaya Gonzalez et al. (2025) demonstrated that constrained sampling can significantly improve seed generation for program fuzzers (Böhme et al., 2016; Herrera et al., 2021). Fuzzers randomly mutate an initial set of input program seeds to generate test cases that trigger different execution paths in a binary. By using grammars to prevent malformed inputs, Anaya Gonzalez et al. (2025) showed that the closer the LM's sampling aligns with the constrained distribution, the more execution paths the fuzzer can explore when generating additional inputs from these seeds.

**Benchmarks.** We evaluate on three targets with varying constraint complexity (details in Appendix G.5): **JSON** processing (requires ≥3 key-value pairs with a fixed first pair), **SQL** testing (mandates two `do_test` blocks per `.test` file), and **XML** parsing (requires 1 element declaration with ≥1 `ATTLIST` in `DOCTYPE`). For each target, we consider two conditions: **prompts with grammar** and **prompts without grammar** (details in Appendix G.2).

**Metrics.** We evaluate *sample efficiency* as the number of generations required to produce 100 valid samples. We also report *line coverage*, the number of unique source code lines executed, measured via LLVM instrumentation (llv, 2025). We use AFL++ (Fioraldi et al., 2020) as our fuzzer and run each fuzzing campaign for one hour, across five independent trials. For each technique, we impose a 2,000-sample cap on generation attempts.

**Findings.** Figure 3 reports results on the XML benchmark without grammar in the prompt. RS and ARS fail to produce 100 valid samples within our 2,000-sample budget. CARS achieves the target with only 440 generations while RSFT requires 893—making our approaches the only practically feasible exact methods. This efficiency advantage holds across benchmarks: for JSON without grammar, CARS requires ∼130 generations versus ∼601 for RSFT (Table 8).

Compared to inexact methods, CARS exhibits better KL divergence (Figures 3a and 5). Remarkably, at comparable sample complexity to MCMC (i.e., the vertical line in the plot), CARS shows significant improvements. The improved faithfulness to the constrained distribution is also reflected in line coverage: CARS-generated seeds lead to ∼7,985 lines covered (Figure 3b) compared to ∼7,115 for GCD and ∼7,965 for MCMC. Similar improvements are observed across other benchmarks (Appendix G.6), with CARS yielding a ∼12% improvement in coverage over GCD for JSON generation (both with and without grammar). We note that SQL-with-grammar proved challenging: no exact approach produced 100 valid samples within budget for our Qwen models, and only RSFT succeeded for Llama-3.1-8B-Instruct (Table 8).

*Table 1.* Molecular generation performance across three chemical classes using Llama-3.1-8B-Instruct. Quality metrics show mean $\pm$ standard deviation over 3 trials. Bold indicates best performance. $\uparrow$ indicates higher is better; $\downarrow$ indicates lower is better.

| | Method | Validity $\uparrow$ | Diversity $\uparrow$ | Retro Score $\uparrow$ | Membership $\uparrow$ | Samples to get 100 Valid Molecules $\downarrow$ |
|---|---|---|---|---|---|---|
| Exact | RS | $0.85 \pm 0.12$ | $0.83 \pm 0.06$ | $0.59 \pm 0.14$ | $0.82 \pm 0.12$ | $793 \pm 127$ |
| | ARS | $\mathbf{0.87 \pm 0.09}$ | $0.83 \pm 0.07$ | $0.56 \pm 0.12$ | $\mathbf{0.85 \pm 0.10}$ | $220 \pm 34$ |
| | RSFT | $0.83 \pm 0.15$ | $0.82 \pm 0.06$ | $0.53 \pm 0.11$ | $0.80 \pm 0.14$ | $765 \pm 89$ |
| | CARS | $\mathbf{0.87 \pm 0.09}$ | $\mathbf{0.85 \pm 0.06}$ | $\mathbf{0.60 \pm 0.15}$ | $\mathbf{0.85 \pm 0.09}$ | $\mathbf{183 \pm 28}$ |
| Approx. | GCD | $0.70 \pm 0.16$ | $0.82 \pm 0.05$ | $0.47 \pm 0.14$ | $0.72 \pm 0.13$ | $\mathbf{100 \pm 0}$ |
| | AWRS | $0.80 \pm 0.02$ | $0.83 \pm 0.01$ | $0.49 \pm 0.03$ | $0.77 \pm 0.04$ | $1000 \pm 0$ |
| | MCMC | $0.79 \pm 0.14$ | $0.84 \pm 0.03$ | $0.51 \pm 0.04$ | $0.77 \pm 0.10$ | $1000 \pm 0$ |

**Summary.** CARS is the only exact method that can handle some fuzzing benchmarks *efficiently* and its distributional fidelity translates to gains in downstream line coverage.

### 4.2. Molecular synthesis

In constrained molecular generation (Kusner et al., 2017; Jin et al., 2019; 2020), both *structural validity* and *chemical diversity* across different samples are crucial for exploring chemical space effectively. This task requires producing valid SMILES (Weininger, 1988) strings that satisfy syntactic constraints (balanced parentheses, valid bonding) and semantic constraints (specific functional groups). We test whether CARS can improve sampling efficiency while maintaining high distributional fidelity in this setting.

**Benchmarks.** We evaluate on three structurally distinct molecular classes from prior work (Wang et al., 2023; Guo et al., 2022): Acrylates (32 example molecules), Chain Extenders (11 example molecules), and Isocyanates (11 example molecules). Each class requires both valid SMILES syntax and class-specific functional group constraints (e.g., acrylate `C=CC(=O)O` motifs). We adopt few-shot prompting with all available exemplars per class, and enforce both syntax and class-level constraints through grammars.

**Metrics.** We measure the four quality dimensions that are considered by Wang et al. (2023): (1) *Validity*: parseability via RDKit (RDKit); (2) *Diversity*: average pairwise Tanimoto distance over Morgan fingerprints (Rogers & Hahn, 2010); (3) *Retrosynthesis Score*: synthesizability via RetroStar (Chen et al., 2020); (4) *Membership*: correct classification in target class. For each method, we generate until obtaining 100 unique molecules in the grammar, excluding the example molecules provided in the prompt, subject to a 1000-sample cap. We averaged across three trials.

**Findings.** CARS consistently delivers advantages in both quality and efficiency. Table 1 shows the results. When accounting for standard deviation, CARS and the other exact methods all achieve the highest molecular diversity and validity. However CARS requires $4.3\times$ fewer samples than

RS, and $1.3\times$ fewer samples than ARS. This reduction in wasted computation translates into substantial practical savings for molecular design pipelines.

The KL divergence of CARS is on average $1.5\times$ lower than MCMC and $1.8\times$ lower than AWRS. MCMC shows characteristic convergence behavior, starting with high divergence ($\sim$26) and gradually decreasing toward CARS's level over multiple steps, but never fully reaching the desired distributional accuracy. The differentiating factor is therefore efficiency, where CARS offers noticeable gains.

Approximate methods suffer in most metrics. Per-class breakdowns in Appendix H.5 confirm these trends across all molecular families and language models. We note that the *diversity* metric is a molecule-specific metric and is not the same as adherence to the exact probability distribution.

**Summary.** In molecular synthesis, where both validity and diversity are essential, CARS achieves the best of both worlds: unbiased sampling that preserves chemical diversity, together with large improvements in computational efficiency over standard rejection sampling.

### 4.3. Text-to-SQL Generation

Constrained text-to-SQL generation (Zhong et al., 2017; Yu et al., 2019) requires producing SQL strings that satisfy both syntactic constraints (proper query structure, valid keywords) and semantic constraints (referencing valid table and column names from the schema). In this experiment, our goal is to assess whether exact samples from a constrained distribution are more likely to produce *correct* queries.

**Benchmarks.** We evaluate on the development split of the Spider dataset (Yu et al., 2019), containing 1,034 examples across 200 databases with varying complexity. Each example consists of a natural language question paired with its corresponding database schema. We present the results for Llama-3.1-8B-Instruct in a zero-shot setting (Appendix I shows similar results for other models), and enforce SQL syntax through a context-free grammar specified in Lark.

*Table 2.* Text-to-SQL generation performance on Spider development set using Llama-3.1-8B-Instruct. Results show mean $\pm$ standard deviation over 4 trials. Bold indicates best performance. $\uparrow$ indicates higher is better; $\downarrow$ indicates lower is better.

| | Method | Execution Accuracy $\uparrow$ | Total Samples $\downarrow$ | Samples/ Query $\downarrow$ |
|---|---|---|---|---|
| Exact | RS | $0.576 \pm 0.014$ | $2126 \pm 155$ | 2.06 |
| | ARS | $0.574 \pm 0.011$ | $1435 \pm 124$ | 1.39 |
| | RSFT | $0.573 \pm 0.009$ | $1916 \pm 186$ | 1.86 |
| | CARS | $\mathbf{0.578 \pm 0.013}$ | $\mathbf{1146 \pm 93}$ | **1.11** |
| Approx. | GCD | $0.525 \pm 0.011$ | $\mathbf{1034 \pm 0}$ | **1.00** |
| | AWRS | $0.567 \pm 0.015$ | $10340 \pm 0$ | 10.00 |
| | MCMC | $0.569 \pm 0.014$ | $10340 \pm 0$ | 10.00 |

**Metrics.** We measure downstream performance via *execution accuracy*, i.e., a binary outcome denoting whether the generated SQL query produces the same results as the ground-truth query when executed on the test database. This metric captures both syntactic validity and semantic correctness. We report results averaged across four trials.

**Findings.** Table 2 shows that exact methods achieve the highest execution accuracy, and CARS has superior sample efficiency among them. Among exact methods, CARS requires $1.85\times$ fewer samples than RS and $1.28\times$ fewer than ARS. Among approximate methods, MCMC and AWRS achieve similar accuracy to the exact methods, but require $9\times$ more samples than CARS; CARS outperforms GCD's accuracy by $5.3\%$, while requiring only $1.1\times$ samples. The KL divergence of CARS is on average $3.2\times$ lower than GCD's, $2.01\times$ lower than MCMC's, and $2.07\times$ lower than AWRS's.

**Summary.** In text-to-SQL generation, CARS provides the best combination of accuracy and efficiency, achieving the highest execution accuracy while requiring fewer samples than both standard rejection sampling methods and approximate techniques that rely on hyperparameter tuning.

### 4.4. Discussion

**When CARS Helps Most.** Across the three benchmarks above, two factors determine the size of CARS's gain over ARS: *(i)* how restrictive the grammar is at each position, i.e., the ratio of valid to invalid tokens out of $|\Sigma|$; and *(ii)* sequence length, since per-position error rates compound multiplicatively. CARS provides the largest gains when the unconstrained acceptance rate is low enough that RS struggles—below roughly $50\%$—while the grammar is tight enough that substantial probability mass can be eliminated per position. For example, in our SMILES (Section 4.2) and SyGuS (Appendix K) experiments the grammar typically permits only 1–20 tokens out of a vocabulary of 128K+, yielding $3.3$–$7\times$ improvement over ARS. Gains are more

modest when the grammar is loose and the LM is already well-aligned with the constraint (e.g., text-to-SQL on Spider, Section 4.3): in that regime ARS itself already has a high acceptance rate, so little additional probability mass remains for CARS to prune.

**When CARS Struggles.** The primary failure mode for CARS is when the space of invalid inputs is composed of many independent low-probability sub-spaces that do not share a prefix. For instance, a constraint that requires a specific token at position 20 can only be falsified after 19 unconstrained tokens, so each rejected sequence eliminates only a tiny amount of probability mass—invalid sequences branch independently rather than sharing prunable prefixes. This is closely related to LM mismatch: when the model assigns negligible probability to the valid set of sequences, even aggressive prefix pruning cannot compensate. In such settings, approximate methods (Anaya Gonzalez et al., 2025) may be the more practical choice; we provide an empirical illustration of this regime in Appendix G.5.

**What the Trie Captures.** Inspecting the trie after a sampling run is informative: the tries stores precisely the constraint information that the LM itself is most prone to violate. For example, on a representative SyGuS task with Llama-3.1-8B-Instruct, the LM places $27.7\%$ of its root-level mass on the token (set and $8.0\%$ on (check—both valid SY-GUS commands in the prompt context, but neither admissible as the start of a body of a solution. CARS removes $128,255$ of $128,256$ tokens at that first position with a single grammar query; ARS requires $128,255$ to remove such tokens. Deeper in the trie, the LM consistently places 2–8% of its probability mass on structurally invalid continuations (e.g., confusing parameter names with non-terminals); these small per-position errors compound multiplicatively over a sequence, which is why CARS's prefix-level pruning yields the $3.3$–$7\times$ gain over ARS observed above. Additional qualitative examples appear in Appendix L.

Operationally, this prefix-level structure also makes the trie cheap to maintain: across our benchmarks $\sim 70\%$ of per-token grammar queries are served from the trie cache rather than re-queried (Appendix F), and trie growth is markedly sublinear—$10\times$ more samples yields only $\sim 3\times$ more nodes on SMILES (Appendix F.6).

## 5. Related Work

**Exact Methods** Our work is a direct improvement of ARS (Gilks & Wild, 2018; Mansinghka et al., 2009), which adaptively rejects previously observed invalid samples. CARS extends ARS to adaptively reject *all* constraint-violating prefix continuations discovered during one sample. Our work shares conceptual foundations with Tromble &

Eisner (2006), who patch a language model with constraints as violations are discovered during decoding. However, their method targets argmax decoding over weighted finite state automatas (FSAs), whereas we focus on sampling from arbitrary autoregressive LMs using "dynamically generated" prefix constraints.

For highly regular constraints (e.g., finite automata), one might hope for a dynamic-programming-style exact sampler that conditions exactly on future feasibility. Such an approach, however, requires tractable marginals over future continuations (i.e., suffix probabilities under the LM), which autoregressive LMs do not expose. Consequently, even when the constraint language is regular, exact constrained sampling still reduces to reasoning about prefixes at the token level—which is precisely what CARS does.

**Static Approximation Methods.** Constrained decoding methods (Scholak et al., 2021; Beurer-Kellner et al., 2023; Geng et al., 2023; Melcer et al., 2024a) enforce constraints incrementally during token-by-token generation. When the constraint is a context-free grammar, this approach is often called Grammar-Constrained Decoding. While efficient at producing valid sequences, these methods modify the LM's probability distribution, resulting in biased samples (Park et al., 2024).

IterGen (Ugare et al., 2025) is a framework for programming constrained decoding algorithms that can move both forward and backward (i.e., backtrack) during generation. While IterGen enables the construction of a wide range of sampling strategies, it does not provide mechanisms for enforcing distributional correctness of the resulting procedures. In contrast, CARS is a concrete sampling algorithm that preserves exactness by construction while achieving efficiency through adaptive pruning. The two approaches therefore target complementary and largely orthogonal objectives.

Gradient-based constrained decoding (Amini et al., 2024; Kumar et al., 2022) similarly steers generation toward satisfying soft or semantic constraints, but cannot guarantee validity (or faithfulness to the distribution) and is computationally expensive.

**Asymptotic Approximation Methods.** Adaptive Sampling with Approximate expected futures (ASAp) (Park et al., 2024) approximates grammar-aligned sampling by building an iterative overapproximation of the probability mass associated with invalid prefixes identified from previous samples. While in the limit this approach reaches the desired distribution, it does not do so *monotonically*—i.e., it can produce intermediate approximations that are very skewed and take thousands of samples to converge to the target distribution. CARS draws inspiration from ASAp in how it maintains and updates a trie of constraint-violating

prefixes. What causes ASAp to distort the distribution is that each sample is produced via constrained decoding, whereas in CARS each sample is produced via rejection sampling.

Monte Carlo techniques, including sequential Monte Carlo (SMC) approaches (Lew et al., 2023; Anaya Gonzalez et al., 2025), sample from constrained distributions by generating multiple candidates using variants of constrained decoding (the static method) and selecting valid ones or resampling partial sequences. AWRS (Lipkin et al., 2025) uses adaptive weighted rejection sampling as a proposal distribution within SMC, tracking rejection statistics to compute importance weights and resampling particles to avoid dead ends. These methods converge to the constrained distribution in the limit, but have no principled stopping criterion, and can be inefficient, requiring nontrivial extra compute.

Other approaches combine LMs with probabilistic models to enforce constraints, e.g., GeLaTo (Zhang et al., 2023) or Ctrl-G (Zhang et al., 2024), often using FSAs or Hidden Markov Models. These methods use surrogate models, are restricted to specific constraint classes, require additional training, and cannot guarantee exact sampling. Approximate inference methods such as Feynman–Kac Transformers (Qin et al., 2022; Lew et al., 2023) share similar limitations.

**Auxiliary-Accelerated Exact Inference.** CARS fits within a broader pattern in statistics and machine learning: using cheap auxiliary signals to accelerate exact statistical procedures without sacrificing their formal guarantees. In CARS, the trie acts as an auxiliary data structure that supplies an increasingly informative proposal distribution, reducing the number of expensive LM forward passes required while preserving exact sampling from $P^{\mathcal{L}}$. The same conceptual move appears, for example, in prediction-powered inference (Ao et al., 2026), where cheap ML predictions are used to reduce labeled sample requirements while still providing valid coverage. CARS can be viewed as an instantiation of this principle in the constrained generation setting, where the auxiliary signal is the prefix-validity oracle and the exact procedure is rejection sampling.

## 6. Conclusion

We introduced *Constrained Adaptive Rejection Sampling* (CARS), a principled extension of Adaptive Rejection Sampling for constrained decoding. Unlike prior methods that either rely on inefficient rejection sampling or approximate the target distribution via MCMC-style procedures, CARS always produces samples from the exact constrained distribution while adaptively pruning entire families of invalid continuations. This combination of fidelity and efficiency makes CARS particularly well-suited for applications where generating diverse, constraint-satisfying samples is critical— e.g., program fuzzing and molecular discovery.

## Acknowledgments

This work was supported in part by a Microsoft Faculty Fellowship; a UCSD JSOE Scholarship; Google's Gemma Academic Program GCP Credit Award; the Kościuszko Foundation, the American Centre of Polish Culture; and NSF under grants CF-2146151, CCF-2546822, CCF-2506134, CCF-2446711, and CCF-2422214l and Schmidt.

Any opinions, findings, and conclusions or recommendations expressed in this publication are those of the authors, and do not necessarily reflect the views of the sponsoring entities. Loris D'Antoni holds concurrent appointments as a Professor at the University of California San Diego and as a Code Metal Scholar. This paper describes work performed at the University of California San Diego and is not associated with Code Metal.

## Impact Statement

This paper presents work whose goal is to advance the field of Machine Learning. There are many potential societal consequences of our work, none which we feel must be specifically highlighted here.

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

# Appendix

Complete experimental details, additional results, and implementation specifics are provided in the appendix:

- **Setup:** Hardware/software specifications (Appendix B), hyperparameters (Appendix C), and model checkpoints (Appendix D)

- **Theory:** Proof of diminishing returns (Appendix E) and computational overhead analysis (Appendix F)

- **Experiments:** Extended results for fuzzing (Appendix G), molecular synthesis (Appendix H), text-to-SQL (Appendix I), PDDL planning (Appendix J), and SyGuS (Appendix K)

## A. Declaration of LLM Usage

Large Language Models (LLMs) are the object of study in this work. However, no LLM was used as a component of our core proposed methodology, or for any part of the experimental data analysis. We used ChatGPT as a writing assistant throughout the research process. Its use included refining prose, generating explanatory text for concepts, drafting document outlines, creating figure captions, and assisting with the generation of boilerplate code for data processing and plotting. All final claims, experimental designs, results, and conclusions were conceived and verified by the human authors, who take full responsibility for the scientific content of this paper.

## B. Hardware and Software

Our experiments were conducted on Ubuntu 22.04 LTS nodes with Intel Xeon Gold 6230 CPUs (2.10 GHz, 10 cores, 20 threads allocated) and 384 GB RAM. For GPU-accelerated workloads, we provisioned a single NVIDIA RTX A6000 GPU for the 7–14B models, and $2\times$ NVIDIA H100 GPUs for the 70B model. Our implementation is based on Python 3.10.12, PyTorch 2.8.0 with CUDA 12.8, an optimized inference backend, and llguidance 0.7.30. For domain-specific experiments, we additionally used: AFL++ 4.00c, LLVM 14.0.0 for fuzzing, RDKit 2025.3.6 for molecular validity checking (SMILES), the Spider evaluation harness (Yu et al., 2019), for text-to-SQL execution accuracy, pyperplan 2.1 and Validate V4 for PDDL planning.

## C. Hyperparameters

For all language model decoding, we set the temperature to 1.0, top-p to 1.0, and top-k to 0 to allow sampling from the full token vocabulary without distributional distortion. We set the maximum number of newly generated tokens as follows:

- **Program fuzzing**: 512 tokens (JSON, XML, SQL)

- **Molecular generation (SMILES)**: 256 tokens

- **Text-to-SQL**: 512 tokens

- **PDDL planning**: 128 tokens (Blocksworld), 256 tokens (Satellite), 1024 tokens (Depot)

- **SyGuS Benchmarks by Park et al. (2024)**: 512 tokens

## D. Model Checkpoint

We evaluate on four instruction-tuned models representing different architectural families and scales:

- **Llama-8B**: Llama-3.1-8B-Instruct (Grattafiori et al., 2024): `https://huggingface.co/meta-llama/Llama-3.1-8B-Instruct` (commit `0e9e39f`)

- **Llama-70B**: Llama-3.1-70B-Instruct (Grattafiori et al., 2024): `https://huggingface.co/meta-llama/Llama-3.1-70B-Instruct` (commit `1605565`)

- **Qwen-7B**: Qwen2.5-7B-Instruct (Qwen et al., 2025): `https://huggingface.co/Qwen/Qwen2.5-7B-Ins` `truct` (commit `a09a354`)

- **Qwen-14B**: Qwen2.5-14B-Instruct (Qwen et al., 2025): `https://huggingface.co/Qwen/Qwen2.5-14B` `-Instruct` (commit `cf98f3b`)

All models use BF16 precision with their default tokenizers. For brevity, we use the short names (Llama-8B, Llama-70B, Qwen-7B, Qwen-14B) in tables and figures throughout the appendix.

## E. Proofs

**Theorem 3.2** (Soundness and Monotonicity). *The CARS algorithm samples an element of $\mathcal{L}$ according to the target distribution $P^{\mathcal{L}}$. Moreover, the adaptive updates performed in Line 10 of the algorithm monotonically increase the probability that some sequence is yielded in Line 8 at subsequent loop iterations.*

*Proof.* Whenever a sequence is produced by the algorithm in Line 5, it comes from the distribution $R^{\mathcal{W}}$, restricted to sequences in $\mathcal{L}$. But in $R^{\mathcal{W}}$ the probability of each sequence $w \in \mathcal{L}$ is proportional to $P(w)$, and the same also holds for $P^{\mathcal{L}}$. The probability that a fixed sequence $w \in \mathcal{L}$ is produced by the algorithm equals $R^{\mathcal{W}}(w) = \frac{P(w)}{p_\epsilon}$. While $P(w)$ is a constant probability, the number $p_\epsilon$ monotonically decreases whenever we add a new invalid prefix to $\mathcal{W}$, causing that $R^{\mathcal{W}}(w)$ increases. The probability that some sequence is produced is just a sum of $R^{\mathcal{W}}(w)$ over all sequences $w \in \mathcal{L}$, hence it increases as well. $\qquad\square$

**Theorem 3.3** (Diminishing Returns). *Let $X_k$ be the random variable saying how much probability mass is removed when the algorithm samples a sequence for the $k$-th time (i.e., it is the value subtracted from $p_\varepsilon$ in that step). Then $\mathbb{E}X_k$ is non-increasing when $k$ grows.*

*Proof.* It is enough to prove that $\mathbb{E}X_1 \geq \mathbb{E}X_2$. This allows us to deduce $\mathbb{E}X_k \geq \mathbb{E}X_{k+1}$ for every $k$ by simply averaging over all possible histories before the $k$-th step (those two steps behave in the same way as the first two steps, assuming a different initial distribution). Then we can conclude with $\mathbb{E}X_k \geq \mathbb{E}X_m$ for $k < m$ by a straightforward induction.

In order to prove $\mathbb{E}X_1 \geq \mathbb{E}X_2$, consider two sequences, $v$ and $w$. Let $d_v$ and $d_w$ be the probabilities of drawing $v$ and $w$, respectively, according to the original distribution. Let $r_v$ be the probability mass removed only while drawing $v$ but not $w$, and likewise let $r_w$ be the probability mass removed only while drawing $w$ but not $v$; finally, let $c$ be the probability mass removed while drawing any of $v$ and $w$. In other words, drawing $v$ as the first sequence removes $r_v + c$, but drawing $v$ after $w$ removes only $r_v$, because the $c$ part was removed already while drawing $w$.

The probability of drawing $v$ first, and then $w$ is

$$d_v \cdot \frac{d_w}{1 - r_v - c},$$

because while drawing the second sequence, the probability mass of $r_v + c$ is already removed, and the probabilities are renormalized. If this event happened, we remove $r_v + c$ in the first step, and $r_w$ in the second step, so this contributes

$$d_v \cdot \frac{d_w}{1 - r_v - c} \cdot (r_v + c)$$

to the expectation $\mathbb{E}X_1$, and

$$d_v \cdot \frac{d_w}{1 - r_v - c} \cdot r_w$$

to the expectation $\mathbb{E}X_2$. In other words, every of those expectations is a sum of the above values over all pairs $v, w$ of distinct sequences. Strictly speaking, it may also happen that the same sequence is sampled twice; this may add something to the expectation $\mathbb{E}X_1$, but necessarily 0 to the expectation $\mathbb{E}X_2$ (no probability mass is removed when obtaining the same sequence again), so in this case the inequality to be shown is trivial.

At the same time, consider the symmetric event, when we draw $w$ first, and then $v$. By symmetry, it contributes

$$d_w \cdot \frac{d_v}{1 - r_w - c} \cdot (r_w + c)$$

to the expectation $\mathbb{E}X_1$, and

$$d_w \cdot \frac{d_v}{1 - r_w - c} \cdot r_v$$

to the expectation $\mathbb{E}X_2$. It suffices to show that the sum of the values added to $\mathbb{E}X_1$, minus the sum of the values added to $\mathbb{E}X_2$ is non-negative, that is,

$$\frac{d_v \cdot d_w \cdot (r_v + c - r_w)}{1 - r_v - c} + \frac{d_w \cdot d_v \cdot (r_w + c - r_v)}{1 - r_w - c} \geq 0.$$

Bringing the two fractions to a common denominator, and removing common factors we obtain

$$(r_v + c - r_w) \cdot (1 - r_w - c) + (r_w + c - r_v)(1 - r_v - c) \geq 0.$$

Expanding the parentheses, reducing repeated components, and grouping the remaining components, we can write this as

$$(r_v - r_w)^2 + c \cdot (2 - r_v - r_w - 2c) \geq 0.$$

Note that $r_v + c \leq 1$ and $r_w + c \leq 1$, because we add probabilities of disjoint events. Thus the above inequality holds, which finishes the proof. □

The above theorem suggests that if we have removed only a small amount of probability mass in multiple recent steps, then it can be expected that no significant amount of probability mass will be removed in future steps. In this situation freezing the trie (and then sampling from the frozen distribution) seems to be a reasonable approach.

## F. Computational Overhead Analysis

We provide an analysis of CARS's computational overhead by profiling across all benchmark tasks. We demonstrate that CARS's trie-based tracking mechanism incurs minimal overhead.

### F.1. Profiling Methodology

We conducted profiling across 37 benchmark runs spanning five domains (program fuzzing, SMILES generation, Text-To-SQL, PDDL planning, and SyGuS), collecting 4,000 successful samples over 24 hours of total runtime. For each run, we tracked:

- Wall-clock time breakdown by operation type

- Memory usage for trie storage (CPU and GPU)

- Trie statistics (nodes, depth, branching factor, reuse rate)

- Operation counts per sample

All profiling was conducted on the hardware described in Appendix B using a custom `Profiler` class that instruments CARS's key operations:

- **Inference timing:** Measured via `time.time()` wrapped around model forward passes

- **Trie operations:** Trie lookups, insertions, and recomputations with per-operation timing

- **Memory profiling:** CPU memory via `psutil`, GPU memory via `torch.cuda.memory_allocated()`

- **Trie structure:** Recursive traversal to compute nodes and their depth

*Table 3.* Runtime breakdown per successful sample across 4,000 samples, using an optimized inference backend.

| Operation | Time (s) | Percentage |
|---|---|---|
| LM Inference | ~8,072 | ~76% |
| Constraint Checking | ~1,912 | ~18% |
| Trie Operations (CARS-specific) | ~425 | ~4% |
| Other (I/O, logging, etc.) | ~213 | ~2% |
| **Total** | **~10,622** | **100%** |

## F.2. Runtime Breakdown

Numbers reported below reflect our current implementation, which uses an optimized inference backend; this yields roughly an $8\times$ wall-clock speedup per sample over our earlier implementation (whose breakdown is preserved in the footnote[1] for reference). Across the same set of runs:

Table 3 shows the average time allocation per successful sample across all benchmarks. We observe,

- **LM inference** dominates at ~76%, inherent to all sampling-based methods.

- **Constraint checking** (~18%) includes probability reweighting and vocabulary masking, required by all constrained decoding methods (GCD, MCMC, CARS).

- **Trie operations** account for ~4% of total runtime under the optimized backend. The relative share of trie operations rises from the earlier $0.3\%$ purely because LM inference is now much faster; the absolute trie cost remains small and CARS's overhead remains minimal.

The trie operations contributing to CARS's trie overhead consist of:

- **Trie lookups:** $O(1)$ amortized per token

- **Trie insertions:** $O(|w||\Sigma|)$ per sample for new prefixes

- **Probability propagation:** $O(|w||\Sigma|)$ per sample for updating parent nodes

Here $\Sigma$ is the set of tokens.

## F.3. Memory Usage

Table 4 summarizes trie memory consumption across benchmarks.

*Table 4.* Trie memory usage statistics across 36 benchmark runs.

| Statistic | Memory (MB) |
|---|---|
| Median | 431 |
| Mean | 8,542 |
| Min | 169 |
| Max | 285,013 |
| 75th percentile | 2,250 |
| 90th percentile | 12,500 |

The maximum memory usage (285GB) occured in the PDDL Satellite domain. For the remaining 39 runs, memory ranged from 169MB to 41GB with median 431MB.

In the worst case, memory usage is proportional to the total length of sampled sequences, since each new sequence contributes its novel suffix (the portion not already in the trie) as new invalid prefix information. In the current implementation, the

---

[1]Earlier (slower) breakdown: LLM Inference 67.3%, Constraint Checking 29.6%, Trie Operations 0.3%, Other 3.1%; ~14 seconds per successful sample.

constant factor is relatively large: to each created trie node we just attach the bit mask saying which tokens can correctly extend the current sequence, as returned by the constraint verifier (parser). One can optimize the memory usage by instead storing a list of tokens correctly extending the current sequence, possibly only in nodes where this list is short (if this list is long, storing it will not save memory anyway, and in this case using a bit mask would be faster). We leave such a memory optimization as a possible future work.

**Freezing the trie.** For memory-constrained applications, users can freeze the trie after collecting a finite number of samples, $N$, and continue sampling from the frozen $R^{\mathcal{W}}$ distribution. This approach bounds the memory while still providing exact sampling with practical rejection rates. Theorem 3.3 provides theoretical justification: the expected probability mass removed per sample is non-increasing, meaning early samples contribute the most to reducing rejection rates. For example, freezing after 100 samples in the PDDL satelite domain case would cap memory at $\sim$2.5GB while maintaining 70%+ of the eventual acceptance rate improvement it would get. In practice, most of the samples that are "helpful" in reducing the rejection rate are discovered in early steps.

### F.4. Trie Reuse Statistics

CARS achieves a 70.4% average trie reuse rate (Table 5), meaning 70.4% of token decisions reuse cached constraint computations rather than querying the constraint checker. This demonstrates that the trie effectively amortizes constraint-checking costs across samples.

*Table 5.* Trie reuse statistics showing percentage of token decisions using cached results.

| Metric | Reuse Rate (%) |
|---|---|
| Mean | 70.4 |
| Median | 76.5 |
| Min | 23.1 |
| Max | 94.2 |

### F.5. Comparison with Baseline Methods

The trie overhead is CARS-specific and represents the algorithmic cost of achieving exact sampling with monotonically improving efficiency. In comparison,

- **RS**: No additional overhead beyond LLM inference and constraint checking, but wastes computation on repeated invalid sequences

- **ARS**: Maintains a hash set of rejected prefixes with O(1) lookup overhead per token (negligible), but still higher rejection rates than CARS as it only learns from complete rejected sequences

- **RSFT**: Maintains a hash set of length-1 invalid prefixes with O(1) lookup overhead (negligible), but only prevents invalid first tokens

- **GCD**: No trie overhead, but still needs to compute token masks; produces biased samples by greedily masking invalid tokens

- **AWRS**: Combines ARS overhead with SMC particle management (tracking importance weights, resampling particles)

- **MCMC**: No trie overhead, but requires multiple Metropolis-Hastings correction steps per sample

The net result is that CARS's small trie overhead enables a 2–10$\times$ reduction in total samples needed (see Section 4), leading to substantial overall speedups despite the marginal computational cost.

### F.6. Trie Memory Analysis

We complement the runtime analysis with an empirical look at how much memory the CARS trie consumes as more samples are drawn. Each trie node stores two vectors of size $|\Sigma|$—the raw next-token log-probabilities from the LM and the adjustment vector $\log \theta$ used to define $R^{\mathcal{W}}$—costing $2|\Sigma|$ floats per node. A node is created only when a new (previously

unseen) prefix is visited, which itself requires a forward pass; thus the trie is bounded by $N \times T$ nodes (where $N$ is the number of sampling attempts and $T$ is the maximum sequence length), but in practice prefix sharing across samples makes the actual node count much smaller.

*Table 6.* Empirical trie size with Llama-3.1-8B-Instruct ($|\Sigma| = 128{,}256$, $\sim751$ KB per node). "Max" and "Avg" are over benchmark instances in the respective domain.

| Benchmark | Samples | Max nodes | Avg nodes | Avg memory |
|---|---|---|---|---|
| SMILES | 100 | 561 | 198 | $\sim145$ MB |
| SMILES | 1000 | 1,858 | 513 | $\sim376$ MB |
| SyGuS (SLIA) | 100 | 5,664 | 832 | $\sim611$ MB |
| SyGuS (SLIA) | 1000 | 14,164 | 2,841 | $\sim2.1$ GB |

Table 6 reports the empirical trie size with Llama-3.1-8B-Instruct. Trie growth is markedly sublinear in the number of samples: a $10\times$ increase in the sample budget yields only a $2.5$–$3.4\times$ increase in average trie size, because later samples mostly traverse cached nodes rather than allocating new ones. This matches the diminishing-returns behavior formalized in Theorem 3.3.

Two practical observations further reduce the effective per-node cost:

- Most entries in the adjustment vector $\log \theta$ are $-\infty$ (corresponding to tokens already known to lead to invalid prefixes); a sparse representation for these vectors substantially reduces per-node memory.

- Once the acceptance rate has stabilized—i.e., once the diminishing-returns regime is reached—the trie can be frozen and low-visit leaf nodes pruned without affecting exactness for the remaining sampling budget (cf. the freezing discussion above).

## G. Fuzzing Experiments

### G.1. Benchmarks

Table 7 summarizes the libraries, versions, and seed formats for each target. We note that, for our XML benchmark, our grammar targets libxml2's DOCTYPE/DTD parsing functionality, representing approximately 10-15% of the library's overall codebase.

*Table 7.* Fuzzing benchmarks, versions, and seed formats.

| Target | Library | Version | Seed format |
|---|---|---|---|
| XML[Group, 2008; 2009] | libxml2 | 2.15.0 | `.xml` |
| SQL[zxteloiv, 2025] | sqlite | 3.50.4 | `.test` |
| JSON[jso] | json-c | 0.18 | `.json` |

### G.2. Prompts and Constraints

For all benchmarks, we use a standard in-context learning format where the prompt consists of two (specification, solution) pairs, followed by a new specification for which the model must generate a solution. A representative prompt for the XML benchmark is shown in Figure 4a. In the "Prompts with Grammar" condition, this same prompt is augmented with the formal grammar specification shown in Figure 4b, while in the "Prompts without Grammar" condition, only the prompt examples are provided.

### G.3. Fuzzing Protocol and Environment

All fuzzing experiments were conducted using AFL++ 4.00c on the hardware and software setup described in Appendix B. Each (benchmark, method) pair was evaluated in $N = 5$ independent, single-instance AFL++ runs of exactly $3600$ s (one hour). We set 'AFL_RANDOM_SEED' to $42 + i, (i = 1...5)$ for reproducibility and configure standard environment variables to ensure non-interactive execution. All other AFL++ parameters remained at defaults to isolate the impact of seed corpus quality. Complete builds and execution scripts are provided in the supplementary materials.

```
You are an expert XML generator.
Make sure you generate valid and diverse XML.

Question 1:
Generate a short, valid and complex XML file.

Solution 1:
<?xml version="1.0" encoding="UTF-8"?>
<!DOCTYPE note [
  <!ELEMENT note (#PCDATA)>
]>
...

Question 2:
Generate a short, valid and complex XML file.

Solution 2:
<?xml version="1.0" encoding="UTF-8"?>
<!DOCTYPE status [
...

Question 3:
Generate a short, valid and complex XML file.

Solution 3:
```

*(a)* Prompt

```
document ::=
    PROLOG doctype_decl element

PROLOG ::=
    "<?xml" attribute* "?>"

doctype_decl ::=
    "<!DOCTYPE" NAME internal_dtd ">"

internal_dtd ::=
    "[" element_decl+ attlist_decl+ "]"

element_decl ::=
    "<!ELEMENT" NAME content_spec ">"

...

attribute ::=
    NAME "=" ESCAPED_STRING

content ::=
    (element | TEXT | cdata)*

cdata ::=
    "<![CDATA[" any_text "]]>"
```

*(b)* Grammar

*Figure 4.* (a) Prompt given to a LM to generate seed test cases for fuzzing the XML parser. (b) Simplified version of the XML grammar written in Lark notation. The goal of the problem is to generate multiple diverse seeds that trigger different code paths in the library being tested.

### G.4. Coverage Measurement via LLVM Instrumentation

We measured line coverage using LLVM's instrumentation toolchain with flags `-fprofile-instr-generate` `-fcoverage-mapping`, which adds $\leq 2\%$ runtime overhead. Raw profiles were collected during execution and aggregated post-trial using `llvm-profdata` and `llvm-cov`.

**Rationale.** We report line coverage rather than crash counts because the experiment isolates *seed quality*—all methods receive identical prompts per benchmark, making coverage a direct measure of how effectively their generated seeds exercise the target code.

### G.5. Complex Case

We document a severe distributional misalignment scenario where even improved rejection sampling methods face fundamental limitations. This case occurs when the grammar constraints mandate syntactic elements that are absent from both the LM's training distribution and the prompt context.

**Experimental Setup.** Initially, we tested SQL constraints in line with those used by Anaya Gonzalez et al. (2025), requiring mandatory `set ::timeout 60000` directives in every `.test` file. This syntax is severely misaligned with typical SQL in LM training data.

**Results.** Across 2000 attempted samples using prompts *without* relevant examples:

- **Standard Rejection Sampling**: $0\%$ acceptance rate.

- **CARS**: $< 0.1\%$ acceptance rate, always times out before reaching 100 valid samples.

This experiment shows that if the LLM is completely misaligned with the target constraint, our approach will not necessarily help. This phenomenon is an expected limitation of rejection sampling and in such settings one should opt for an inexact approach. To enable meaningful comparison between exact and approximate methods, we instead use the SQLite test-script grammar with mandatory `do_test` blocks—a challenging but feasible benchmark where exact methods remain viable.

## G.6. Results

This section provides comprehensive fuzzing results across all benchmarks and conditions, complementing the representative results shown in Section 4.1. We present results for three grammar-intensive targets (JSON, SQL, XML) across three models (Llama-8B, Qwen-7B, Qwen-14B; see Appendix D) under both prompt conditions (with/without grammar specification).

**AWRS Infeasibility.** We mark AWRS as computationally infeasible (INFE.) in Table 8 for all fuzzing benchmarks. AWRS is a Sequential Monte Carlo (SMC) method that maintains $M$ particles, each requiring its own forward pass during resampling steps. With $M = 10$ particles (the setting from the original paper), the memory and compute requirements exceeded the capacity of our hardware (Appendix B) when combined with the complex grammars used in our fuzzing benchmarks (e.g., Figure 4b). We were unable to complete a single generation within reasonable time and memory limits, and thus exclude AWRS from these experiments.

**Sample Efficiency Summary.** Table 8 shows the number of LM generations required to produce 100 valid samples across all experimental conditions. Methods that timeout within the 2000-sample budget are marked as such.

*Table 8.* Sample efficiency across fuzzing benchmarks—generations required for 100 valid samples. T.O. indicates timeout (2000-sample budget exceeded). INFE. indicates computationally infeasible.

| Method | JSON | | | SQL | | | XML | | |
|---|---|---|---|---|---|---|---|---|---|
| | Llama-8B | Qwen-7B | Qwen-14B | Llama-8B | Qwen-7B | Qwen-14B | Llama-8B | Qwen-7B | Qwen-14B |
| | *Without Grammar in Prompt* | | | | | | | | |
| RS | T.O. | T.O. | T.O. | T.O. | T.O. | T.O. | T.O. | T.O. | T.O. |
| ARS | T.O. | T.O. | T.O. | T.O. | T.O. | T.O. | T.O. | T.O. | T.O. |
| RSFT | ~601 | ~341 | ~319 | ~1960 | T.O. | ~1953 | ~893 | ~1601 | ~1453 |
| CARS | ~130 | ~230 | ~197 | ~1004 | ~1240 | ~1189 | ~440 | ~442 | ~398 |
| GCD | 100 | 100 | 100 | 100 | 100 | 100 | 100 | 100 | 100 |
| AWRS | INFE. | INFE. | INFE. | INFE. | INFE. | INFE. | INFE. | INFE. | INFE. |
| MCMC | 1000 | 1000 | 1000 | 1000 | 1000 | 1000 | 1000 | 1000 | 1000 |
| | *With Grammar in Prompt* | | | | | | | | |
| RS | T.O. | T.O. | T.O. | T.O. | T.O. | T.O. | T.O. | T.O. | T.O. |
| ARS | T.O. | T.O. | T.O. | T.O. | T.O. | T.O. | ~413 | ~612 | ~601 |
| RSFT | ~874 | ~127 | ~124 | ~1560 | T.O. | T.O. | ~275 | ~731 | ~704 |
| CARS | ~475 | ~131 | ~121 | T.O. | T.O. | ~1883 | ~215 | ~548 | ~398 |
| GCD | 100 | 100 | 100 | 100 | 100 | 100 | 100 | 100 | 100 |
| AWRS | INFE. | INFE. | INFE. | INFE. | INFE. | INFE. | INFE. | INFE. | INFE. |
| MCMC | 1000 | 1000 | 1000 | 1000 | 1000 | 1000 | 1000 | 1000 | 1000 |

**Line Coverage Results.** Tables 9, 10, and 11 show downstream fuzzing performance measured by line coverage achieved after 1 hour of AFL++ execution.

*Table 9.* Line coverage results for JSON fuzzing benchmarks. Values show mean lines covered $\pm$ 95% CI over 5 independent trials.

| Method | Without Grammar | | | With Grammar | | |
|---|---|---|---|---|---|---|
| | Llama-8B | Qwen-7B | Qwen-14B | Llama-8B | Qwen-7B | Qwen-14B |
| RS | T.O. | T.O. | T.O. | T.O. | T.O. | T.O. |
| ARS | T.O. | T.O. | T.O. | T.O. | T.O. | T.O. |
| RSFT | $3,120 \pm 40$ | $3,050 \pm 30$ | $3,110 \pm 40$ | $3,080 \pm 50$ | $3,060 \pm 40$ | $3,140 \pm 40$ |
| CARS | $3,230 \pm 50$ | $3,090 \pm 40$ | $3,200 \pm 50$ | $3,180 \pm 60$ | $3,090 \pm 30$ | $3,240 \pm 30$ |
| GCD | $2,870 \pm 30$ | $2,850 \pm 20$ | $2,940 \pm 50$ | $2,890 \pm 40$ | $2,860 \pm 30$ | $2,950 \pm 50$ |
| AWRS | INFE. | INFE. | INFE. | INFE. | INFE. | INFE. |
| MCMC | $3,100 \pm 40$ | $3,190 \pm 50$ | $3,220 \pm 40$ | $3,150 \pm 40$ | $3,170 \pm 60$ | $3,240 \pm 50$ |

**KL Divergence.** Figure 5 shows distributional fidelity across benchmarks and conditions, measured as KL divergence from the empirically estimated target distribution.

*Table 10.* Line coverage results for SQL fuzzing benchmarks. Values show mean lines covered $\pm$ 95% CI over 5 independent trials.

| Method | Without Grammar | | | With Grammar | | |
|---|---|---|---|---|---|---|
| | Llama-8B | Qwen-7B | Qwen-14B | Llama-8B | Qwen-7B | Qwen-14B |
| RS | T.O. | T.O. | T.O. | T.O. | T.O. | T.O. |
| ARS | T.O. | T.O. | T.O. | T.O. | T.O. | T.O. |
| RSFT | $22{,}223 \pm 394$ | T.O. | $22{,}875 \pm 267$ | $20{,}174 \pm 473$ | T.O. | T.O. |
| CARS | $22{,}456 \pm 315$ | $21{,}278 \pm 473$ | $22{,}827 \pm 254$ | T.O. | T.O. | $22{,}855 \pm 237$ |
| GCD | $20{,}726 \pm 236$ | $20{,}016 \pm 236$ | $20{,}142 \pm 148$ | $19{,}700 \pm 315$ | $20{,}488 \pm 394$ | $20{,}871 \pm 311$ |
| AWRS | INFE. | INFE. | INFE. | INFE. | INFE. | INFE. |
| MCMC | $21{,}041 \pm 315$ | $21{,}435 \pm 394$ | $21{,}631 \pm 322$ | $19{,}858 \pm 236$ | $21{,}320 \pm 315$ | $21{,}783 \pm 289$ |

*Table 11.* Line coverage results for XML fuzzing benchmarks. Values show mean lines covered $\pm$ 95% CI over 5 independent trials.

| Method | Without Grammar | | | With Grammar | | |
|---|---|---|---|---|---|---|
| | Llama-8B | Qwen-7B | Qwen-14B | Llama-8B | Qwen-7B | Qwen-14B |
| RS | T.O. | T.O. | T.O. | T.O. | T.O. | T.O. |
| ARS | T.O. | T.O. | T.O. | $7{,}871 \pm 219$ | $7{,}891 \pm 178$ | $8{,}014 \pm 241$ |
| RSFT | $7{,}966 \pm 356$ | $7{,}942 \pm 442$ | $8{,}061 \pm 387$ | $7{,}944 \pm 247$ | $8{,}033 \pm 351$ | $8{,}041 \pm 312$ |
| CARS | $7{,}984 \pm 264$ | $8{,}045 \pm 326$ | $8{,}041 \pm 339$ | $8{,}033 \pm 445$ | $8{,}022 \pm 267$ | $8{,}039 \pm 267$ |
| GCD | $7{,}117 \pm 178$ | $7{,}209 \pm 267$ | $7{,}249 \pm 231$ | $7{,}844 \pm 356$ | $7{,}120 \pm 178$ | $7{,}609 \pm 198$ |
| AWRS | INFE. | INFE. | INFE. | INFE. | INFE. | INFE. |
| MCMC | $7{,}964 \pm 356$ | $7{,}977 \pm 267$ | $8{,}011 \pm 118$ | $7{,}933 \pm 178$ | $7{,}937 \pm 415$ | $8{,}023 \pm 215$ |

**Key Findings.** The comprehensive results reveal several important patterns across our three fuzzing benchmarks,

- **Method Feasibility and Timeout Patterns**—Rejection sampling methods (RS, ARS) consistently timeout across all benchmarks and conditions, confirming the computational intractability of naive approaches for constrained generation. AWRS proves to be computationally INFE. for all tested scenarios, for the infrastructure used in Section B highlighting the limitations of existing weighted approaches for complex constraint satisfaction.

- **CARS Performance Superiority**—Where feasible, CARS achieves the highest line coverage across most conditions. For JSON benchmarks, CARS reaches 32.3% coverage without grammar (vs. 28.7% for GCD), representing a 12.5% improvement. In XML fuzzing, CARS consistently achieves 9.7-9.8% coverage, outperforming all baselines including MCMC's 9.3-9.7% range.

- **Distributional Fidelity Translates to Coverage Quality**—The superior downstream fuzzing performance of CARS-generated seeds demonstrates that maintaining distributional fidelity under constraints yields tangible benefits in exploration diversity. Across benchmarks, CARS consistently outperforms approximate methods like GCD by 3-4%, confirming that exact sampling methods provide meaningful advantages for seed generation in fuzzing applications.

# H. Molecular Generation (SMILES)

## H.1. Experimental Setup

We evaluate on three molecular classes from prior work (Wang et al., 2023; Guo et al., 2022), representing distinct industrial chemical applications:

**Acrylates** (32 example molecules): Vinyl ester compounds for polymer synthesis, characterized by the `C=CC(=O)O` motif. **Chain Extenders** (11 example molecules): Difunctional molecules for polymer chain extension with hydroxyl or amine groups. **Isocyanates** (11 example molecules): Reactive compounds with `N=C=O` groups for polyurethane synthesis.

Each class employs hierarchical SMILES grammars enforcing both syntactic validity (balanced parentheses, valid bonds, ring closures) and semantic constraints (required functional groups). Figure 7 illustrates the prompt structure and grammar for acrylates; similar constructions apply to other classes.

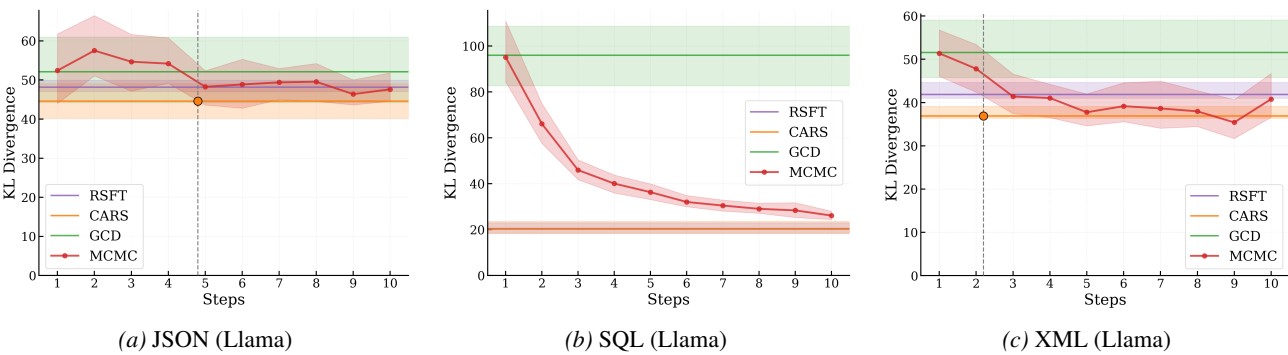

*Figure 5.* KL divergence comparison across fuzzing benchmarks (without grammar condition). CARS and RSFT show consistently lower divergence than approximate methods, confirming distributional fidelity while MCMC shows convergence behavior over steps.

## H.2. Parse-Tree Illustration

Figure 6 illustrates the parse tree for a representative Acrylate molecule (`C=CC(=O)OCC`). The tree demonstrates how the grammar enforces both SMILES syntax and the required acrylate functional group. Purple nodes represent non-terminals, green nodes show grammar terminals, and blue text displays the actual SMILES tokens.

## H.3. Prompts and Constraints

We use few-shot prompting where all available exemplars for each class serve as context. Figure 7a shows the prompt structure for Acrylates. The model receives all 32 known acrylates as examples, then must generate novel molecules satisfying the grammar in 7b.

## H.4. Evaluation Metrics

We assess four complementary aspects of molecular generation quality,

**Validity.** Fraction of generated SMILES successfully parsed by RDKit (RDKit) without errors. This measures basic chemical plausibility.

**Diversity.** Average pairwise Tanimoto distance computed over Morgan fingerprints (Rogers & Hahn, 2010) with radius 2 and 2048 bits:

$$D = \frac{2}{n(n-1)} \sum_{i<j} (1 - T(M_i, M_j))$$

where $T$ is Tanimoto similarity and $M_i$ are molecular fingerprints. Higher values indicate more diverse chemical space coverage.

**Retrosynthesis Score.** Synthesizability estimated via RetroStar (Chen et al., 2020), which predicts reaction pathways to available building blocks. Scores range [0,1] with higher values indicating easier synthesis.

**Class Membership.** Fraction correctly classified into the target chemical class via SMARTS pattern matching for required functional groups.

**Sample Efficiency.** Mean number of LM forward passes required to obtain 100 valid, unique molecules (excluding prompt exemplars). We impose a 1000-sample timeout and average over 3 independent trials.

**Note on drug-likeness metrics.** We intentionally omit QED (Bickerton et al., 2012) and Lipinski's Rule of Five (Lipinski et al., 2012) as the industrial chemical classes used in our evaluation (polymers, coatings) are not intended for pharmaceutical applications. Such metrics would be inappropriate for evaluating polymer precursors and specialty chemicals.

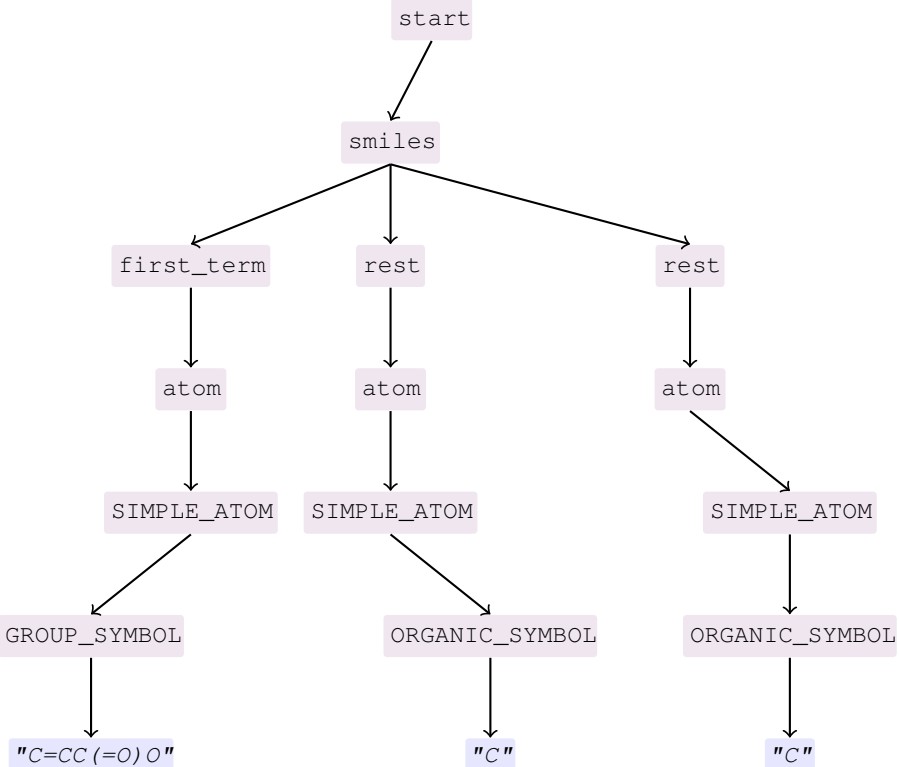

*Figure 6.* Parse tree for ethyl acrylate (`C=CC(=O)OCC`). The tree shows how grammar constraints enforce the acrylate functional group (highlighted) while permitting variation in alkyl substituents. Purple nodes represent non-terminals, and *blue italics* text displays the actual SMILES tokens.

```
You are an expert in chemistry. You
are given several examples of acrylates
molecules in SMILES format. Your task is
to provide one new, valid acrylates
molecule in the SMILES format. Your
response must be a single SMILES
molecule and nothing else.

Molecule: C=CC(=O)OCC1=CC=CC=C1
Molecule: C=CC(=O)OC1=CC=CC=C1
Molecule: CC(=C)C(=O)OC1=CC=CC=C1
Molecule: C=CC(=O)OCCC1=CC=CC=C1
Molecule: CCCCCCCCOC(=O)C(=C)C
Molecule: CCC(C)OC(=O)C=C
Molecule: CC(=C)C(=O)OCC1=CC=CC=C1
Molecule: CCCOC(=O)C=C
Molecule: CC(C)COC(=O)C(=C)C
Molecule: CCCCCCCCCCCCOC(=O)C(=C)C
Molecule: CCC(C)OC(=O)C(=C)C
... (16 molecules)
Molecule: CCCOC(=O)C(=C)C
Molecule: CC1CC(CC(C1)(C)C)OC(=O)C(=C)C
Molecule: CC(C)CCCCCCCOC(=O)C=C
Molecule: CCCOC(=O)C=C
Molecule: COCCOC(=O)C=C
Molecule:
```

*(a)* Prompt

```
start : smiles

smiles : first_term rest*

first_term : atom branch* RING_CLOSURE*

rest : BOND? (atom branch* RING_CLOSURE* |
RING_CLOSURE)
...

SIMPLE_ATOM :
  ORGANIC_SYMBOL | AROMATIC_SYMBOL |
  WILDCARD | GROUP_SYMBOL

BOND : "-" | "=" | "#" | "$" | ":" |
    "/" | "\" | "."

ORGANIC_SYMBOL : "Br" | "Cl" | "N" | "O" |
    "P" | "S" | ...

AROMATIC_SYMBOL : "b" | "c" | "n" | "o" | ...

GROUP_SYMBOL : "C=CC(=O)O" | ...
...

ISOTOPE : "1".."9" ("0".."9")? ("0".."9")?
```

*(b)* Grammar

*Figure 7.* Acrylate generation setup showing (a) few-shot prompting with all 32 class exemplars and (b) simplified grammar enforcing both SMILES syntax and acrylate functional groups.

## H.5. Results

Tables 12–14 show detailed performance breakdown by chemical class across all three models (see Appendix D for model details). Chain Extenders show the highest baseline validity rates; Isocyanates present the most challenging generation task.

*Table 12.* Molecular generation results for Acrylates. Values show mean $\pm$ standard deviation over 3 trials. * indicates less than 100 valid samples were used for the evaluation. More details in Table 15.

| Model | Method | Validity ↑ | Diversity ↑ | Retro Score ↑ | Membership ↑ |
|---|---|---|---|---|---|
| Llama-8B | RS* | $0.88 \pm 0.01$ | $0.76 \pm 0.01$ | $0.73 \pm 0.02$ | $0.88 \pm 0.01$ |
| | ARS* | $0.91 \pm 0.01$ | $0.74 \pm 0.00$ | $0.75 \pm 0.04$ | $0.90 \pm 0.01$ |
| | RSFT | $0.87 \pm 0.04$ | $0.74 \pm 0.02$ | $0.66 \pm 0.04$ | $0.86 \pm 0.04$ |
| | CARS | $0.93 \pm 0.03$ | $0.81 \pm 0.01$ | $0.71 \pm 0.04$ | $0.92 \pm 0.02$ |
| | GCD | $0.61 \pm 0.01$ | $0.76 \pm 0.01$ | $0.50 \pm 0.01$ | $0.69 \pm 0.01$ |
| | AWRS | $0.60 \pm 0.01$ | $0.78 \pm 0.01$ | $0.50 \pm 0.01$ | $0.70 \pm 0.02$ |
| | MCMC | $0.57 \pm 0.02$ | $0.78 \pm 0.02$ | $0.51 \pm 0.01$ | $0.71 \pm 0.02$ |
| Qwen-7B | RS* | $0.92 \pm 0.01$ | $0.29 \pm 0.00$ | $0.98 \pm 0.02$ | $0.98 \pm 0.01$ |
| | ARS* | $1.00 \pm 0.00$ | $0.23 \pm 0.01$ | $0.99 \pm 0.00$ | $1.00 \pm 0.00$ |
| | RSFT | $1.00 \pm 0.00$ | $0.16 \pm 0.03$ | $1.00 \pm 0.00$ | $1.00 \pm 0.00$ |
| | CARS | $0.99 \pm 0.00$ | $0.25 \pm 0.02$ | $0.98 \pm 0.01$ | $0.99 \pm 0.00$ |
| | GCD | $0.70 \pm 0.01$ | $0.17 \pm 0.02$ | $0.60 \pm 0.03$ | $0.70 \pm 0.02$ |
| | AWRS | $0.79 \pm 0.01$ | $0.27 \pm 0.02$ | $0.77 \pm 0.03$ | $0.78 \pm 0.04$ |
| | MCMC | $0.78 \pm 0.01$ | $0.25 \pm 0.01$ | $0.78 \pm 0.01$ | $0.78 \pm 0.02$ |
| Qwen-14B | RS* | $0.94 \pm 0.01$ | $0.26 \pm 0.01$ | $0.98 \pm 0.00$ | $0.99 \pm 0.01$ |
| | ARS* | $1.00 \pm 0.00$ | $0.25 \pm 0.02$ | $0.99 \pm 0.00$ | $1.00 \pm 0.00$ |
| | RSFT | $1.00 \pm 0.00$ | $0.22 \pm 0.03$ | $1.00 \pm 0.00$ | $1.00 \pm 0.00$ |
| | CARS | $0.99 \pm 0.00$ | $0.27 \pm 0.03$ | $0.98 \pm 0.01$ | $0.99 \pm 0.00$ |
| | GCD | $0.73 \pm 0.02$ | $0.22 \pm 0.02$ | $0.68 \pm 0.03$ | $0.69 \pm 0.02$ |
| | AWRS | $0.81 \pm 0.01$ | $0.30 \pm 0.03$ | $0.79 \pm 0.03$ | $0.78 \pm 0.04$ |
| | MCMC | $0.82 \pm 0.02$ | $0.27 \pm 0.02$ | $0.81 \pm 0.02$ | $0.80 \pm 0.02$ |

**Sample Efficiency Analysis.** Table 15 provides detailed sample efficiency breakdown, showing the number of generations required to produce 100 valid molecules across models and chemical classes.

**Key Findings.**

- **Acrylates**: CARS achieves best validity (0.93) and diversity (0.81) on Llama-8B, with competitive performance across all models.

- **Chain Extenders**: Highest baseline validity across methods; CARS maintains competitive performance.

- **Isocyanates**: Most challenging class. CARS and ARS tie for best validity (0.76) on Llama-8B.

## I. Text-to-SQL Generation

Table 16 presents text-to-SQL generation results on the Spider development set across all three models (see Appendix D for model details). Exact methods consistently achieve the highest execution accuracy while CARS maintains superior sample efficiency.

**Key Findings.** Qwen-7B demonstrates better overall performance on Spider compared to Llama-8B, with CARS achieving 0.593 execution accuracy (vs. 0.578 for Llama-8B). The relative improvements remain consistent across models:

- CARS outperforms GCD by 5.2–5.4% points in execution accuracy

- CARS achieves higher accuracy than MCMC and AWRS while requiring $\sim 9\times$ fewer samples

*Table 13.* Molecular generation results for Chain Extenders. Values show mean $\pm$ standard deviation over 3 trials.

| Model | Method | Validity ↑ | Diversity ↑ | Retro Score ↑ | Membership ↑ |
|---|---|---|---|---|---|
| Llama-8B | RS | $0.95 \pm 0.02$ | $0.87 \pm 0.01$ | $0.58 \pm 0.02$ | $0.90 \pm 0.02$ |
| | ARS | $0.94 \pm 0.01$ | $0.87 \pm 0.00$ | $0.52 \pm 0.06$ | $0.91 \pm 0.01$ |
| | RSFT | $0.94 \pm 0.02$ | $0.87 \pm 0.01$ | $0.53 \pm 0.01$ | $0.91 \pm 0.04$ |
| | CARS | $0.93 \pm 0.00$ | $0.88 \pm 0.01$ | $0.54 \pm 0.03$ | $0.91 \pm 0.01$ |
| | GCD | $0.91 \pm 0.00$ | $0.86 \pm 0.01$ | $0.57 \pm 0.01$ | $0.89 \pm 0.00$ |
| | AWRS | $0.93 \pm 0.00$ | $0.87 \pm 0.01$ | $0.53 \pm 0.00$ | $0.03 \pm 0.00$ |
| | MCMC | $0.92 \pm 0.00$ | $0.87 \pm 0.00$ | $0.58 \pm 0.01$ | $0.90 \pm 0.00$ |
| Qwen-7B | RS | $0.98 \pm 0.00$ | $0.76 \pm 0.01$ | $0.31 \pm 0.02$ | $0.98 \pm 0.00$ |
| | ARS | $0.99 \pm 0.00$ | $0.76 \pm 0.02$ | $0.38 \pm 0.04$ | $0.99 \pm 0.00$ |
| | RSFT | $0.99 \pm 0.00$ | $0.75 \pm 0.03$ | $0.31 \pm 0.01$ | $0.99 \pm 0.00$ |
| | CARS | $0.99 \pm 0.00$ | $0.75 \pm 0.02$ | $0.37 \pm 0.01$ | $0.99 \pm 0.00$ |
| | GCD | $1.00 \pm 0.00$ | $0.71 \pm 0.04$ | $0.22 \pm 0.03$ | $0.97 \pm 0.01$ |
| | AWRS | $1.00 \pm 0.00$ | $0.77 \pm 0.01$ | $0.28 \pm 0.02$ | $0.97 \pm 0.00$ |
| | MCMC | $1.00 \pm 0.00$ | $0.75 \pm 0.03$ | $0.30 \pm 0.04$ | $0.98 \pm 0.01$ |
| Qwen-14B | RS | $0.99 \pm 0.00$ | $0.77 \pm 0.02$ | $0.32 \pm 0.01$ | $0.98 \pm 0.01$ |
| | ARS | $1.00 \pm 0.00$ | $0.77 \pm 0.03$ | $0.38 \pm 0.03$ | $0.99 \pm 0.00$ |
| | RSFT | $1.00 \pm 0.00$ | $0.78 \pm 0.01$ | $0.33 \pm 0.02$ | $0.99 \pm 0.00$ |
| | CARS | $1.00 \pm 0.00$ | $0.78 \pm 0.01$ | $0.38 \pm 0.02$ | $1.00 \pm 0.00$ |
| | GCD | $1.00 \pm 0.00$ | $0.72 \pm 0.03$ | $0.25 \pm 0.04$ | $0.97 \pm 0.01$ |
| | AWRS | $1.00 \pm 0.00$ | $0.78 \pm 0.01$ | $0.31 \pm 0.01$ | $0.98 \pm 0.00$ |
| | MCMC | $1.00 \pm 0.00$ | $0.75 \pm 0.03$ | $0.30 \pm 0.03$ | $0.98 \pm 0.01$ |

- Among exact methods, CARS uses 1.8–1.9× fewer samples than RS and 1.3× fewer than ARS while maintaining the highest accuracy

## J. PDDL Planning

In this experiment, we consider a benchmark where the goal is not to sample many diverse outputs but to solve a concrete task. Our goal is to assess whether exact samples from a constrained distribution are more likely to solve a downstream task.

We evaluate on three PDDL (Planning Domain Definition Language) settings from Zuo et al. (2025); Wang et al. (2023): *Blocks World*, *Depot*, and *Satellite*. For each domain, we construct few-shot prompts using four ground-truth plans and test on four randomly sampled tasks, targeting 100 valid action plans with a 1000-sample cap. Results are averaged over three independent trials.

### J.1. Benchmarks

We evaluate on three classical planning domains from (Zuo et al., 2025; Wang et al., 2023):

- **Blocks World** (4 tasks): Stacking and unstacking blocks to achieve goal configurations

- **Depot** (4 tasks): Logistics domain with trucks, hoists, and crates requiring coordinated movement and loading operations across multiple locations.

- **Satellite** (4 tasks): Satellite observation scheduling with actions for pointing instruments, calibrating, and taking images of celestial targets.

Each domain employs PDDL action grammars that enforce:

- **Syntactic validity**: Correct PDDL action syntax with proper operator names, parameter lists, and parenthesis matching.

- **Type constraints**: Parameters must match declared object types (e.g., `block`, `truck`, `satellite`).

- **Arity constraints**: Correct number of arguments for each action operator.

*Table 14.* Molecular generation results for Isocyanates. Values show mean $\pm$ standard deviation over 3 trials.

| Model | Method | Validity ↑ | Diversity ↑ | Retro Score ↑ | Membership ↑ |
|---|---|---|---|---|---|
| Llama-8B | RS | $0.72 \pm 0.09$ | $0.87 \pm 0.01$ | $0.45 \pm 0.08$ | $0.68 \pm 0.08$ |
| | ARS | $0.76 \pm 0.06$ | $0.86 \pm 0.00$ | $0.48 \pm 0.01$ | $0.71 \pm 0.06$ |
| | RSFT | $0.65 \pm 0.04$ | $0.86 \pm 0.00$ | $0.41 \pm 0.01$ | $0.63 \pm 0.04$ |
| | CARS | $0.76 \pm 0.01$ | $0.87 \pm 0.01$ | $0.47 \pm 0.02$ | $0.72 \pm 0.03$ |
| | GCD | $0.64 \pm 0.00$ | $0.87 \pm 0.00$ | $0.32 \pm 0.00$ | $0.64 \pm 0.01$ |
| | AWRS | $0.65 \pm 0.01$ | $0.84 \pm 0.02$ | $0.36 \pm 0.01$ | $0.66 \pm 0.01$ |
| | MCMC | $0.67 \pm 0.00$ | $0.86 \pm 0.00$ | $0.37 \pm 0.00$ | $0.67 \pm 0.02$ |
| Qwen-7B | RS | $0.87 \pm 0.04$ | $0.76 \pm 0.03$ | $0.41 \pm 0.04$ | $0.87 \pm 0.03$ |
| | ARS | $0.86 \pm 0.03$ | $0.76 \pm 0.02$ | $0.41 \pm 0.03$ | $0.86 \pm 0.02$ |
| | RSFT | $0.91 \pm 0.02$ | $0.77 \pm 0.01$ | $0.42 \pm 0.03$ | $0.91 \pm 0.02$ |
| | CARS | $0.87 \pm 0.02$ | $0.76 \pm 0.03$ | $0.43 \pm 0.02$ | $0.87 \pm 0.02$ |
| | GCD | $0.90 \pm 0.02$ | $0.79 \pm 0.01$ | $0.58 \pm 0.01$ | $0.89 \pm 0.02$ |
| | AWRS | $0.91 \pm 0.01$ | $0.80 \pm 0.01$ | $0.49 \pm 0.02$ | $0.88 \pm 0.03$ |
| | MCMC | $0.90 \pm 0.02$ | $0.79 \pm 0.01$ | $0.51 \pm 0.01$ | $0.90 \pm 0.02$ |
| Qwen-14B | RS | $0.89 \pm 0.02$ | $0.78 \pm 0.01$ | $0.42 \pm 0.03$ | $0.87 \pm 0.04$ |
| | ARS | $0.88 \pm 0.03$ | $0.76 \pm 0.02$ | $0.44 \pm 0.02$ | $0.86 \pm 0.01$ |
| | RSFT | $0.91 \pm 0.02$ | $0.77 \pm 0.01$ | $0.43 \pm 0.04$ | $0.91 \pm 0.03$ |
| | CARS | $0.90 \pm 0.02$ | $0.78 \pm 0.01$ | $0.44 \pm 0.01$ | $0.90 \pm 0.02$ |
| | GCD | $0.88 \pm 0.02$ | $0.77 \pm 0.01$ | $0.53 \pm 0.01$ | $0.88 \pm 0.01$ |
| | AWRS | $0.92 \pm 0.02$ | $0.78 \pm 0.01$ | $0.54 \pm 0.01$ | $0.90 \pm 0.05$ |
| | MCMC | $0.91 \pm 0.02$ | $0.78 \pm 0.01$ | $0.53 \pm 0.02$ | $0.88 \pm 0.03$ |

*Table 15.* Sample efficiency for molecular generation: generations required for 100 valid molecules. * indicates extrapolation when fewer than 100 valid samples were produced.

| | Llama-8B | | | Qwen-7B | | | Qwen-14B | | |
|---|---|---|---|---|---|---|---|---|---|
| Method | Acry. | Chain | Iso. | Acry. | Chain | Iso. | Acry. | Chain | Iso. |
| RS | 2100* | 105 | 139 | 3000* | 102 | 115 | 2960* | 101 | 117 |
| ARS | 871 | 106 | 132 | 129 | 101 | 116 | 125 | 101 | 113 |
| RSFT | 2100* | 106 | 154 | 3333* | 101 | 110 | 2988* | 101 | 108 |
| CARS | 277 | 108 | 132 | 112 | 101 | 115 | 109 | 101 | 106 |
| GCD | 100 | 100 | 100 | 100 | 100 | 100 | 100 | 100 | 100 |
| AWRS | 1000 | 1000 | 1000 | 1000 | 1000 | 1000 | 1000 | 1000 | 1000 |
| MCMC | 1000 | 1000 | 1000 | 1000 | 1000 | 1000 | 1000 | 1000 | 1000 |

Figure 8b shows the grammar for Satellite actions. The grammar ensures syntactic correctness but does not enforce semantic constraints (preconditions/effects), which are verified separately during evaluation.

## J.2. Prompts and Constraints

We use four-shot in-context learning where each example contains a PDDL problem specification (initial state and goal) paired with its ground truth action plan. The prompt includes the domain specification, which defines the available actions and object types. Figure 8a shows the prompt structure for Satellite.

## J.3. Evaluation Metrics

Following (Zuo et al., 2025; Loula et al., 2025), we employ evaluation metrics with increasing orders of strictness,

**Sample Efficiency.** Mean number of LM forward passes required to obtain 100 parseable plans. We impose a 1000-LM-generation timeout per task and average over 4 tasks for every domain, across 3 trials.

**Prefix Validity.** Among parseable plans, this is the fraction of plans where the first 4 actions are: (1) Executable from the initial state (preconditions are satisfied);(2) Result in a state from which the goal is reachable (verified via search). This

*Table 16.* Text-to-SQL generation performance on Spider development set. Quality metrics show mean $\pm$ standard deviation over 4 trials.

| Model | Method | Execution Accuracy ↑ | Total Samples ↓ | Samples/Query ↓ |
|---|---|---|---|---|
| Llama-8B | RS | $0.576 \pm 0.014$ | $2126 \pm 155$ | $\sim 2.06$ |
| | ARS | $0.574 \pm 0.011$ | $1435 \pm 124$ | $\sim 1.39$ |
| | RSFT | $0.573 \pm 0.009$ | $1916 \pm 186$ | $\sim 1.86$ |
| | CARS | $0.578 \pm 0.013$ | $1146 \pm 93$ | $\sim 1.11$ |
| | GCD | $0.525 \pm 0.011$ | $1034 \pm 0$ | $1.00$ |
| | AWRS | $0.567 \pm 0.015$ | $10340 \pm 0$ | $10.0$ |
| | MCMC | $0.569 \pm 0.014$ | $10340 \pm 0$ | $10.0$ |
| Qwen-7B | RS | $0.593 \pm 0.012$ | $2047 \pm 142$ | $\sim 1.98$ |
| | ARS | $0.591 \pm 0.013$ | $1389 \pm 117$ | $\sim 1.34$ |
| | RSFT | $0.589 \pm 0.010$ | $1852 \pm 163$ | $\sim 1.79$ |
| | CARS | $0.593 \pm 0.011$ | $1108 \pm 87$ | $\sim 1.07$ |
| | GCD | $0.541 \pm 0.009$ | $1034 \pm 0$ | $1.00$ |
| | AWRS | $0.582 \pm 0.016$ | $10340 \pm 0$ | $10.0$ |
| | MCMC | $0.584 \pm 0.013$ | $10340 \pm 0$ | $10.0$ |
| Qwen-14B | RS | $0.601 \pm 0.011$ | $1931 \pm 121$ | $\sim 1.86$ |
| | ARS | $0.611 \pm 0.015$ | $1297 \pm 87$ | $\sim 1.25$ |
| | RSFT | $0.613 \pm 0.011$ | $1782 \pm 111$ | $\sim 1.72$ |
| | CARS | $0.614 \pm 0.013$ | $1094 \pm 42$ | $\sim 1.06$ |
| | GCD | $0.554 \pm 0.010$ | $1034 \pm 0$ | $1.00$ |
| | AWRS | $0.596 \pm 0.014$ | $10340 \pm 0$ | $10.0$ |
| | MCMC | $0.597 \pm 0.012$ | $10340 \pm 0$ | $10.0$ |

metric assesses semantic coherence and planning feasibility.

**Ground Truth Similarity.** Exact match rate between the first 4 generated actions and the reference solution. This measures alignment with expert planning strategies.

**Rationale for metrics.** PDDL generation from natural language is challenging - models frequently produce syntactically correct but semantically invalid plans - especially for problems with over 10 objects. The cascading framework distinguishes surface-level correctness (parsing) from deeper planning competence (executability, goal-directedness).

Natural language–to–PDDL generation is notoriously difficult: models often produce sequences that are syntactically malformed or semantically invalid. For semantic quality, we follow the cascading evaluation by Zuo et al. (2025); Loula et al. (2025) and measure the metrics above.

## J.4. Results

Table 17 summarizes efficiency results across all three models (see Appendix D for model details). The results highlight strong variation in constraint alignment across models: Qwen-7B and Qwen-14B achieve moderate alignment, while Llama-8B fails to produce 100 samples within the cap for RS/ARS.

**Key Findings.**

- **Sample Efficiency**: For Qwen-7B, CARS uses $1.2\times$ fewer LM calls than ARS (the next best exact method). For Llama-8B, RS and ARS fail to produce 100 samples, while 61% of CARS's LM calls produce valid samples.

- **Distributional Fidelity**: The KL divergence of CARS is on average $2.1\times$ lower than MCMC and $2.8\times$ lower than AWRS.

- **Semantic Quality**: Overall low across all methods, reflecting the inherent difficulty of PDDL generation. Nonetheless, exact methods slightly outperform approximate methods on Prefix Validity and Ground Truth Similarity.

```
You are a PDDL planning expert. You are
given a domain, and some examples of
planning problems and a valid sequences
to achieve the goal.
...
Your final output must be a valid
sequence of actions.

Domain: SATELLITE

Domain Definition:
(define (domain satellite)
(:requirements :strips)
(:predicates
  (on_board ?i ?s) ... )
...
Problem:
(:objects
...
)
(:init
 (satellite satellite0)
...
 (direction Phenomenon7)
)
(:goal (and
 (pointing satellite0 Phenomenon5)
... (have_image Star4 spectrograph2)
...
))

Solution:
```

*(a)* Prompt

```
start : PLAN

PLAN :
    ACTION (" " ACTION)*

ACTION :
    "(" action_body ")"

action_body :
    binary_action " " OBJECT ...

binary_action :
    "switch_on" | "switch_off"

ternary_action :
    "turn_to" | "calibrate"

quaternary_action :
    "take_image"

OBJECT :
    "instrument" digit_0_7
    | "satellite" digit
    | "groundstation" digit
    | "phenomenon" digits
    | "planet" digits
    | "star" digits
...

digit_0_7 : "0".."7"
digit : "0".."9"
digits : digit+
```

*(b)* Grammar

*Figure 8.* (a) 4-shot prompt for Satellite planning. (b) Simplified version of the Satellite PDDL actions written in Lark notation. The grammar enforces correct action syntax for satellite manipulation operations.

## K. SyGuS Benchmarks by Park et al. (2024)

For completeness, we also evaluate CARS on the synthesis benchmarks introduced by Park et al. (2024). These tasks involve synthesizing expressions in an extension of linear integer arithmetic (SLIA) and loop invariants with bit-vector arithmetic (BV4). The problems are specified in the Syntax-Guided Synthesis (SyGuS) format (Alur et al., 2019), which provides both a logical specification and a context-free grammar of admissible terms. Following prior work, prompts consist of three in-context examples (specification–solution pairs), and the grammar is then given as a constraint for grammar-aligned sampling. The full benchmark contains 29 problems (14 BV4 and 15 SLIA).

While SyGuS is a natural setting for constrained generation, this benchmark is a somewhat imperfect fit for our problem formulation. The metric of interest here is the ability to produce *many diverse valid samples*, yet in real synthesis applications the key goal is to obtain *a single correct solution*. Thus, although we report results for completeness and comparability with prior work, we view this evaluation as secondary to the benchmarks in the main text.

**Setup.** We compare CARS against three rejection sampling variants (RS, ARS, RSFT) and report three trials per method. Each trial generates 100 samples for each of the 29 problems, with a limit of 10000 LM calls. If the limit of 10000 calls is reached, we report the number of samples produced within that limit.

**Efficiency.** Figure 9 reports the number of model calls required to generate 100 samples (each bar shows the median of 3 runs). Standard rejection sampling (RS) fails completely, often producing zero samples within the timeout. Restricting only the first token (RSFT) already helps substantially, since models otherwise tend to start with phrases like "*The solution is*" rather than a valid program. Still, CARS achieves order-of-magnitude improvements: on BV4, CARS uses $16\times$ fewer calls to the LM (geometric mean) than ARS and $5.7\times$ fewer than RSFT; on SLIA, the corresponding factors are $4.5\times$ and $11.4\times$.

**Summary.** Although the SyGuS benchmarks are not directly aligned with the one-solution synthesis objective that motivates CARS, they nonetheless confirm the central message: *CARS transforms rejection sampling from essentially unusable into a highly efficient constrained generator, outperforming prior rejection-based methods.*

*Table 17.* PDDL Planning results: sample efficiency and semantic quality metrics. In case of a timeout (T.O.), we measure semantic quality on the <100 results produced before the timeout.

| Model | Method | % Valid ↑ | Prefix Validity ↑ | Gr. Truth Similarity ↑ |
|---|---|---|---|---|
| Llama-8B | RS | T.O. | 0.2% | 0.0% |
| | ARS | T.O. | 0.7% | 0.0% |
| | RSFT | 36% | 3.0% | 0.5% |
| | CARS | 61% | 2.7% | 0.5% |
| | GCD | 100% | 1.0% | 0.0% |
| | MCMC | 100% | 0.7% | 0.0% |
| | AWRS | 100% | 1.0% | 0.1% |
| Qwen-7B | RS | 38% | 4.0% | 1.2% |
| | ARS | 54% | 4.3% | 0.9% |
| | RSFT | 51% | 6.4% | 1.9% |
| | CARS | 66% | 6.3% | 2.5% |
| | GCD | 100% | 2.0% | 1.0% |
| | MCMC | 100% | 2.6% | 1.7% |
| | AWRS | 100% | 1.4% | 0.4% |
| Qwen-14B | RS | 41% | 5.8% | 1.9% |
| | ARS | 55% | 6.1% | 1.4% |
| | RSFT | 53% | 6.7% | 2.3% |
| | CARS | 70% | 6.9% | 2.7% |
| | GCD | 100% | 2.1% | 1.0% |
| | MCMC | 100% | 2.9% | 1.7% |
| | AWRS | 100% | 2.5% | 1.3% |

# L. Qualitative Case Study: What CARS Prunes

To make concrete what the trie "learns" over the course of sampling, we walk through two representative cases using Llama-3.1-8B-Instruct: one SyGuS (SLIA) synthesis task and one SMILES generation task. The point of this section is not to add a new quantitative result but to illustrate the *structure* of the probability mass that CARS prunes—which is the source of its gains over ARS.

## L.1. SyGuS (SLIA, name-combine-4_short)

The task is to synthesize a name-formatting function. At the root of the trie, the LM places $27.7\%$ of its mass on the token (set and $8.0\%$ on (check—both valid SYGUS commands in the prompt, but neither is admissible as the start of a solution body. A single grammar query at the root eliminates $128{,}255$ of $128{,}256$ tokens; under ARS, none of this mass would be discovered until an entire $-terminated sample is rejected.

Deeper in the trie, the LM repeatedly confuses parameter names with non-terminals (e.g., placing $2.5\%$ of its mass on nt at a position where only lastname is grammatically valid). These per-position error rates of 2–8% are individually small but compound multiplicatively over the length of the program. CARS converts each such error into a permanent prefix-level prune, while ARS only learns from the corresponding rejected complete sequence; empirically, CARS requires $\sim 7\times$ fewer attempts than ARS on this task.

Over a 1000-sample run, CARS discovered $\sim 764$M invalid (token, position) pairs, of which only 374 originated from ARS-style "learn after rejection" updates. The rest came from prefix-level pruning at positions the sampler had already passed through.

## L.2. SMILES (Acrylate Monomers)

At the root of the trie, the LM spreads roughly $1\%$ of its mass over English-language tokens such as M, There, and Here—reflecting the model's uncertainty about whether to emit a SMILES string or to begin a natural-language explanation. The grammar oracle rejects all such tokens at position 0.

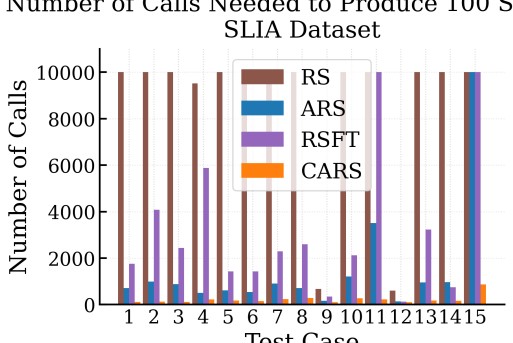

*Figure 9.* Number of of LM calls required to produce 100 samples for BV4 and SLIA. Lower is better.

Mid-molecule, the LM assigns 2–10% of its mass to structurally invalid continuations: opening a branch where the grammar requires the current ring to close, proposing an invalid ring-closure digit, or proposing a carbon chain at a position where only an oxygen is admissible by the acrylate functional-group constraint. CARS learns ∼71.5M invalid (token, position) pairs over the run, leading to ∼3.3× fewer attempts than ARS.

**Takeaway.**  The probability mass that CARS prunes is structural—it depends on the current parse state—not the kind of "easy" surface mistake that a larger model would simply stop making. This is consistent with the larger-model experiments in Appendix M: scaling the LM does not eliminate the constraint-induced waste that CARS is designed to remove.

## M. Larger-Model Experiments (Llama-3.1-70B-Instruct)

To check that the benefits of CARS do not vanish at larger scale, we additionally evaluate Llama-3.1-70B-Instruct on two of our main benchmarks: SMILES (Acrylates) molecular generation and text-to-SQL on the Spider development set. Setup, prompts, and grammars are identical to those used for the smaller models (Appendices H and I).

**Setup.**  Inference for the 70B model is run on 2×NVIDIA H100 GPUs using the same optimized inference backend as the rest of our experiments (Appendix B). All other hyperparameters (temperature, top-$p$, top-$k$, max tokens, sampling budgets) match the corresponding 7–14B configuration.

**Results: Molecular Generation (Acrylates).**  Table 18 reports validity, diversity, retro score, and class membership for Acrylates with the 70B model, alongside sample efficiency (generations to obtain 100 valid molecules). Results are mean ± standard deviation over 3 trials.

*Table 18.* Acrylate molecular generation with Llama-3.1-70B-Instruct. Sample efficiency is the number of generations to reach 100 valid samples (lower is better).

| Method | Validity ↑ | Diversity ↑ | Retro Score ↑ | Membership ↑ | Samples ↓ |
|---|---|---|---|---|---|
| RS | $0.91 \pm 0.02$ | $0.77 \pm 0.01$ | $0.78 \pm 0.02$ | $0.90 \pm 0.02$ | $432 \pm 61$ |
| ARS | $0.94 \pm 0.01$ | $0.76 \pm 0.01$ | $0.80 \pm 0.03$ | $0.93 \pm 0.02$ | $312 \pm 47$ |
| RSFT | $0.90 \pm 0.03$ | $0.75 \pm 0.02$ | $0.70 \pm 0.03$ | $0.89 \pm 0.04$ | $346 \pm 34$ |
| CARS | $0.95 \pm 0.01$ | $0.82 \pm 0.02$ | $0.76 \pm 0.03$ | $0.94 \pm 0.02$ | $198 \pm 24$ |
| GCD | $0.66 \pm 0.01$ | $0.77 \pm 0.02$ | $0.55 \pm 0.01$ | $0.73 \pm 0.01$ | $100 \pm 0$ |
| AWRS | $0.65 \pm 0.01$ | $0.79 \pm 0.02$ | $0.55 \pm 0.01$ | $0.74 \pm 0.02$ | $1000 \pm 0$ |
| MCMC | $0.63 \pm 0.02$ | $0.79 \pm 0.01$ | $0.56 \pm 0.01$ | $0.74 \pm 0.02$ | $1000 \pm 0$ |

**Results: Text-to-SQL (Spider).**  Table 19 reports execution accuracy, total samples to cover all 1,034 Spider development queries, and samples per query for the 70B model. Results are mean ± standard deviation over 4 trials.

*Table 19.* Text-to-SQL execution accuracy and sample efficiency with Llama-3.1-70B-Instruct on the Spider development set.

| Method | Execution Accuracy ↑ | Total Samples ↓ | Samples/Query ↓ |
|---|---|---|---|
| RS | $0.676 \pm 0.012$ | $1193 \pm 43$ | $\sim 1.15$ |
| ARS | $0.678 \pm 0.012$ | $1098 \pm 47$ | $\sim 1.06$ |
| RSFT | $0.677 \pm 0.010$ | $1147 \pm 31$ | $\sim 1.10$ |
| CARS | $0.678 \pm 0.011$ | $1064 \pm 23$ | $\sim 1.03$ |
| GCD | $0.627 \pm 0.010$ | $1034 \pm 0$ | $1.00$ |
| AWRS | $0.665 \pm 0.013$ | $10340 \pm 0$ | $10.0$ |
| MCMC | $0.667 \pm 0.012$ | $10340 \pm 0$ | $10.0$ |

**Discussion.** Structural constraints continue to bind even at the 70B scale: the LM still places non-negligible mass on grammatically invalid continuations, and CARS retains its sample-efficiency advantage over RS and ARS while matching the execution accuracy and downstream quality of the other exact methods. This is consistent with prior observations that scaling alone does not eliminate constraint-induced waste in structured-output generation (Lipkin et al., 2025; Loula et al., 2025).

