# OpenReview forum: "Constrained Adaptive Rejection Sampling"
_ICML.cc/2026/Conference — ICML 2026 regular_

### Official Review · Reviewer_hGpX · 2026-03-04

**Soundness:** 4
**Presentation:** 3
**Significance:** 3
**Originality:** 3
**Overall Recommendation:** 5
**Confidence:** 3

**Summary:**

CARS (Constrained Adaptive Rejection Sampling) is an algorithm for exact sampling from language-model distributions conditioned on structural constraints (valid JSON, SQL, programs, molecules). When a sample is rejected, CARS scans every prefix of the rejected sample and adds all constraint-violating continuations to a trie-based invalid-prefix set — rather than adding just one prefix as in standard ARS. This can substantially increase pruning per rejection. The paper proves soundness (Theorem 3.2: CARS preserves the exact constrained LM distribution) and monotonicity (acceptance probability monotonically increases with each update). Evaluation across grammar-based fuzzing, molecular synthesis, and text-to-SQL with Llama-3.1-8B and Qwen2.5-7B/14B shows significant efficiency gains over RS and ARS while maintaining exactness, and better distributional fidelity than approximate methods (GCD, MCMC, AWRS).

**Compliance With Llm Reviewing Policy:**

Affirmed.

**Final Justification:**

The author fully resolved my concerns and I look forward to the improvement in camera-ready

**Key Questions For Authors:**

1. Can you provide bounds (theoretical or empirical) on trie memory growth?
2. How does CARS interact with speculative decoding or batched inference?
3. For constraints that are only partially prefix-checkable (some violations detectable early, others not), can CARS degrade gracefully?
4. CARS's core idea — exploiting cheap auxiliary signals (prefix checks) to accelerate an exact statistical procedure (rejection sampling) while preserving formal guarantees — resonates with a broader pattern in efficient inference. For instance, prediction-powered inference (arXiv:2601.21470) uses ML predictions to reduce labeled sample requirements while maintaining valid coverage, and best-arm identification with LLM judges (arXiv:2601.21471) combines cheap LLM evaluations with expensive human feedback to reduce query complexity. Have you considered framing CARS within this "auxiliary-accelerated exact inference" perspective? It could help connect the constrained-decoding community with the broader statistical efficiency literature.

**Limitations:**

The prefix-checkability requirement is clearly acknowledged. Memory growth and scaling to larger models are not discussed in sufficient depth.

**Strengths And Weaknesses:**

**Strengths:**

1. The core algorithmic insight is clean and non-obvious. The distinction between ARS (1 invalid prefix per rejection) and CARS (a whole family of invalid prefixes) is sharp, and the idea of squeezing maximal information out of each failed sample is well thought out.

2. The monotonic improvement guarantee (Theorem 3.2) is practically meaningful: acceptance probability increases with each update, unlike ASAp (non-monotonic) or MCMC (asymptotic only). In the XML fuzzing benchmark, RS/ARS fail to produce 100 valid samples in 2,000 attempts while CARS succeeds with ~215 generations — a concrete illustration.

3. The evaluation is well-designed: three diverse constraint domains, baselines covering both exact (RS, ARS) and approximate (GCD, MCMC, AWRS) methods, and a clean ablation across update strategies which isolates each component's contribution. The overhead is negligible (~0.3% of compute).

4. The fuzzing experiments connect distributional fidelity to downstream utility: CARS-generated inputs improve branch coverage by 9.9% vs. 7.2% for GCD and 9.6% for MCMC.

**Weaknesses:**

**Major:**

1. **Constraint class scope.** CARS requires "prefix-checkable" constraints — one must efficiently determine whether a given prefix can be extended to a satisfying string. This covers CFGs, regular languages, and some type systems, but excludes semantic correctness, test-passing code, factual accuracy, or safety properties. This limitation is acknowledged, but a more formal characterization of the boundary (with concrete examples of constraints falling just outside) would help practitioners assess applicability.

2. **No memory analysis for the trie.** The trie grows with each rejection. For tight constraints and long sequences, I worry this could become problematic. Either a theoretical bound on trie size (as a function of rejections, vocabulary size, and constraint complexity), or an empirical plot of memory usage across iterations, would address this gap.

**Minor:**

3. **Marginal gains in easy cases.** For text-to-SQL, CARS needs only 1.11 samples/query vs. 2.06 for RS. For PDDL with Qwen (38% acceptance), gains are modest. I'd appreciate a clearer picture of the "sweet spot" — when does CARS give you qualitative improvement vs. marginal gains?

4. **Model scale.** All experiments use 7B–14B models. For larger models with presumably higher constraint compliance, the relative advantage may diminish. Even a brief argument about how the advantage scales with model capability would be helpful.

---

> ### Author Rebuttal · Authors · 2026-03-31
>
> > **C1.** CARS requires  ... practitioners assess applicability.
>
> > **Q3.** For constraints that are only ... can CARS degrade gracefully?
>
> Thanks for the observation. We ask Reviewer hGpX to see our response to Reviewer pFvs (*C1*). In short, CARS degrades gracefully to ARS when prefix checking is unavailable at a given position.
>
> ---
>
> > **C2.** The trie grows with each rejection ... would address this gap
>
> > **Q1.** Can you provide bounds ... on trie memory growth?
>
> Each trie node stores two vectors of size `|V|` (raw log-probabilities and the adjustment log θ), costing `2|V|` floats per node. A node is created only when a new prefix is visited, which requires a model forward pass. The trie is therefore bounded by the total number of unique prefixes visited: at most `N x T` nodes for `N` attempts with maximum sequence length `T`, though prefix sharing makes the actual count much smaller.
>
> Empirically with Llama-8B (`|V|` = 128,256, ~751 KB/node), we profile memory across two benchmarks,
>
> SMILES: 561 nodes max (412 MB) / 198 avg (145 MB) at 100 samples; 1,858 nodes max (1.4 GB) / 513 avg (376 MB) at 1000 samples.
>
> SyGuS (SLIA): 5,664 nodes max (4.1 GB) / 832 avg (611 MB) at 100 samples; 14,164 nodes max (10.0 GB) / 2,841 avg (2.1 GB) at 1000 samples.
>
> Growth is sublinear - for 10x more samples, trie size grows only 2.5–3.4x on average, as later samples mostly traverse cached nodes. For memory-constrained settings: (1) most entries in log θ are -inf, so a sparse representation would reduce per-node cost; (2) once the acceptance rate stabilizes, the trie can be frozen and low-visit leaf nodes pruned without meaningful loss in acceptance probability.
>
> ---
>
> > **C3.** For text-to-SQL ... qualitative improvement vs. marginal gains?
>
> The benefit of CARS depends on two factors: (1) how restrictive the grammar is at each position (i.e., the ratio of valid to invalid tokens), and (2) the sequence length, since per-position error rates compound multiplicatively. In our qualitative study (see response to Reviewer VCTL (*C1, Q2*), we observe that SMILES and SyGuS grammars frequently permit only 1–20 tokens out of 128,256 at a given position - CARS blocks the remaining ~128K tokens with a single grammar query, preventing the 2–10% per-position invalid mass from compounding across the sequence. For text-to-SQL, the grammar is comparatively loose and the LM is well-aligned with SQL syntax, so the gains are more modest when compared to ARS.
>
> As a practical heuristic: CARS provides the largest gains when the unconstrained acceptance rate is low enough that RS struggles (say, below 50%) and the grammar is tight enough that we can eliminate substantial mass at each position. The 3.3-7x improvement over ARS we observe on SMILES and SyGuS reflects this sweet spot.
>
> ---
>
> > **C4.** All experiments use 7B–14B models ... would be helpful.
>
> This is a great point. The constraints CARS blocks are structural – they depend on the grammar's parse state (open parentheses, ring closures, syntactic position). Findings from [1, 2] confirm that adaptive rejection-based methods remain beneficial even at 70B scale. We can include larger-model results on select benchmarks in the final version.
>
> [1] Loula, J. et al. Syntactic and Semantic Control of Large Language Models via Sequential Monte Carlo. ICLR 2025.
>
> [2] Lipkin, B. et al. Fast Controlled Generation from Language Models with Adaptive Weighted Rejection Sampling. COLM 2025.
>
> ---
>
> > **Q3.** How does CARS ... batched inference?
>
> We thank the reviewer for this question. For speculative decoding, we have not yet explored a direct integration. The main challenge is that CARS modifies the sampling distribution at each token via trie-based reweighting, which would need to be reconciled with the draft-then-verify paradigm. However, we note that the trie naturally provides a complementary form of speedup: as sampling progresses, previously visited prefixes have their full distributions cached, allowing CARS to skip expensive forward passes entirely on revisits.
>
> For batched inference, CARS processes samples sequentially by design – the trie improves with each attempt, so sequential processing maximizes learning from failures (as in Theorem 3.3). That said, batching is fully compatible: samples within a batch can share the same reweighted distribution without intra-batch learning, and the trie is then collectively updated from the entire batch before the next round.
>
> ---
>
> > **Q4.** CARS's core idea ... statistical efficiency literature.
>
> This is a great question and thanks for pointing us to this literature. We are in fact in the process of extending CARS to the setting where, instead of having a hard constraint, our goal is to sample with respect to an arbitrary “scoring function” given by an auxiliary model. Think “exact sampling from a product of experts”. Such an approach would bring the framing closer to what the reviewer pointed us to, where an auxiliary model acts as a soft constraint.

---

> > ### Author Rebuttal · Reviewer_hGpX · 2026-04-02
> >
> > I thank the authors for the response. The trie memory profiling is helpful, especially the sublinear growth numbers and the practical mitigation strategies. The sweet-spot characterization (tight grammar + low acceptance rate) is also a useful addition for practitioners.
> >
> > The product-of-experts extension mentioned in Q4 sounds promising. I would also be happy to see the camera-ready discuss the broader connections I raised in my review, as I think situating CARS within the "auxiliary-accelerated exact inference" perspective could help the paper reach readers beyond the constrained-decoding community.
> >
> > I have also read the responses to other reviewers on the constraint class scope and partial prefix-checkability, which look reasonable. I maintain my score and look forward to the camera-ready with the promised 70B-scale results.

---

> > > ### Author Response · Authors · 2026-04-05
> > >
> > > Thanks! We'll gladly add the connection to the work pointed by the reviewer as well as new experiments.

---

### Official Review · Reviewer_VCTL · 2026-03-10

**Soundness:** 3
**Presentation:** 3
**Significance:** 3
**Originality:** 3
**Overall Recommendation:** 5
**Confidence:** 4

**Summary:**

This paper proposes an exact constrained sampling algorithm that extends Adaptive Rejection Sampling (ARS) by tracking all invalid prefix continuations discovered per sample in a trie and subtracting their probability mass from future samples.

**Compliance With Llm Reviewing Policy:**

Affirmed.

**Final Justification:**

My concerns have been resolved.

**Key Questions For Authors:**

- I wonder if the authors have explored integration into inference frameworks such as vllm or constrained sampling frameworks such as (Loula et al. 25)? It would further speed the method up and enable a fairer comparison in app F--since the authors used Transformers for inference, which is considerably slower than state-of-the-art frameworks.

- A case study would be great to compare the method with previous ones and to see how it improves over them. E.g., what prefixes are ruled out in the algorithm and what are their likelihoods? It would be interesting to see whether such prefixes could be ruled out by language models larger in size without applying the algorithm.

**Limitations:**

yes

**Strengths And Weaknesses:**

Strengths:
- This paper is theoretically sound: it samples from the exact constrained distribution, with monotonically non-decreasing acceptance rates (Theorem 3.2 and 3.3)
- The experiments are solid. Most relevant domains that require constrained sampling are covered.
- I like how the runtime analysis is done in App F, which makes everything very clear. (Please see one question below related to App F)
- The performance of the proposed method is good. It improves sample efficiency by a large margin (Table 1).

Weaknesses:

The only thing I could think of is that some qualitative examples would help explain the advantages of the proposed method. It would be interesting to see what kinds of prefixes get ruled out by the algorithm and to conduct some analysis on them.

---

> ### Author Rebuttal · Authors · 2026-03-31
>
> > **Q1.** I wonder if the authors have explored integration into inference frameworks such as vllm or constrained sampling frameworks such as (Loula et al. 25)? It would further speed the method up and enable a fairer comparison in app F--since the authors used Transformers for inference, which is considerably slower than state-of-the-art frameworks.
>
> We thank the reviewer for this practical suggestion. Since submission, we have implemented a vLLM backend for CARS (using the vLLM V0 engine, similar to [1, 2]), and observed a ~8x wall-clock speedup over our Transformers-based implementation. The updated profiling confirms that CARS's overhead remains minimal (as compared to Appendix F): model inference dominates at ~76% of wall time, constraint checking accounts for ~18%, and trie operations account for roughly 4%. We can include these updated results in the revision after rerunning our full set of experiments.
>
> [1] Loula, J. et al. Syntactic and Semantic Control of Large Language Models via Sequential Monte Carlo. ICLR 2025.
>
> [2] Lipkin, B. et al. Fast Controlled Generation from Language Models with Adaptive Weighted Rejection Sampling. COLM 2025.
>
> ---
> > **C1.** The only thing I could think of is that some qualitative examples would help explain the advantages of the proposed method. It would be interesting to see what kinds of prefixes get ruled out by the algorithm and to conduct some analysis on them.
>
> > **Q2.** A case study would be great to compare the method with previous ones and to see how it improves over them. E.g., what prefixes are ruled out in the algorithm and what are their likelihoods? It would be interesting to see whether such prefixes could be ruled out by language models larger in size without applying the algorithm.
>
> We provide a case study on two tasks from our benchmarks using Llama-3.1-8B-Instruct.
>
> SyGuS (SLIA, name-combine-4_short): Synthesizing a name formatting function (e.g., "John", "Doe" -> "Doe, J."). At the root, the LM assigns 27.7% to `(set` and 8.0% to `(check` – valid SyGuS commands from the prompt, but wrong for a solution. CARS eliminates 128,255/128,256 tokens with a single grammar query. Deeper in the trie, the LM confuses parameter names with nonterminals (2.5% on `nt` where only `lastname` is valid). These per-position error rates of 2-8% compound multiplicatively across the sequence — CARS prevents this, requiring 7x fewer attempts than ARS. In total, CARS discovers 764M invalid (token, position) pairs, of which only 374 came from the rejection-based learning that ARS uses.
>
> SMILES (acrylate monomers): At the root, the LM spreads ~1% mass on English words like "M", "There", "Here" – uncertain whether to produce SMILES or natural language. Mid-molecule, the LM consistently assigns 2-10% mass to structurally invalid continuations – opening branches where the grammar requires closure, proposing invalid ring digits, or suggesting carbon chains where only oxygen is valid. CARS learns 71.5M invalid pairs – and in practice requires 3.3x fewer attempts than ARS.
>
>
> These constraints are structural, i.e., they depend on parse state. Additionally, findings from [1, 2] confirm that adaptive rejection-based methods remain beneficial even at 70B scale. We can include larger-model results on select benchmarks in the final version.
>
> [1] Loula, J. et al. Syntactic and Semantic Control of Large Language Models via Sequential Monte Carlo. ICLR 2025.
>
> [2] Lipkin, B. et al. Fast Controlled Generation from Language Models with Adaptive Weighted Rejection Sampling. COLM 2025.

---

> > ### Author Rebuttal · Reviewer_VCTL · 2026-03-31
> >
> > Thanks for the clarification! Great work! I'll keep my already positive scores.

---

### Official Review · Reviewer_pFvs · 2026-03-13

**Soundness:** 3
**Presentation:** 3
**Significance:** 3
**Originality:** 3
**Overall Recommendation:** 5
**Confidence:** 4

**Summary:**

This paper introduces Constrained Adaptive Rejection Sampling for exact constrained generation from autoregressive language models. The core idea is to retain the distributional correctness of rejection sampling while improving amortized efficiency by learning from failures more aggressively: when a sampled continuation violates the constraint, CARS does not just blacklist that rejected output, but records the shortest invalid prefix and, for each proper prefix encountered, all next-token continuations that are provably invalid. These invalid prefixes are stored in a trie, and their probability mass is subtracted from future draws via a reweighted distribution, so later samples avoid revisiting already-eliminated invalid regions. The paper proves that CARS still samples exactly from the constrained distribution, and that the acceptance probability improves monotonically as more invalid regions are pruned.

**Compliance With Llm Reviewing Policy:**

Affirmed.

**Key Questions For Authors:**

I would appreciate a more direct discussion of how CARS compares to other possible exact approaches that exploit structured constraints, especially in regimes where the constraint language is highly regular and might admit dynamic-programming-style treatment.

Also, could you discuss the failure mode where acceptance remains near zero, e.g., is this mostly driven by LM mismatch to the target language, by weak informativeness of the prefix oracle, or by the fact that many invalid regions are only discovered too late to help much?

**Limitations:**

yes

**Strengths And Weaknesses:**

Strengths:

This paper tackles an important problem, which is how to obtain exact constrained samples from autoregressive LMs without suffering the severe inefficiency of vanilla rejection sampling. The core idea of CARS, namely to cache and reuse information about invalid prefixes in a trie so that future samples avoid already-eliminated invalid regions, is easy to understand, and meaningfully extends prior adaptive rejection approaches. I found the theoretical contribution strong as the exactness guarantee and monotonic improvement in acceptance probability give the method a strong formal foundation, and the paper is also honest about the assumptions under which the method applies. Empirically, the evaluation is broad and convincing and the results consistently support the claim that CARS improves sample efficiency over exact baselines while retaining better fidelity than approximate methods.

Weaknesses:

The main limitation is that the applicability of CARS depends heavily on having an efficient incremental prefix-validity oracle over the token vocabulary, which restricts the method to a narrower class of constraints than the paper’s broad framing might initially suggest. While this is acknowledged, it means the approach may not transfer easily to more semantic or black-box constraints where prefix-level invalidity is hard to detect. I also think some of the empirical claims about fidelity are somewhat indirect, since comparisons for approximate baselines rely on proxies rather than direct distance to the true constrained distribution, and some baselines are infeasible in the settings where CARS looks strongest, which makes certain comparisons less complete than ideal. In addition, although the trie overhead itself is small, it would help clarifying the end to end practical tradeoffs in regimes with weak pruning signal or very low LM mass on the target language, since the appendix suggests CARS can still fail badly there.

---

> ### Author Rebuttal · Authors · 2026-03-31
>
> > **C1.** The main limitation is ... prefix-level invalidity is hard to detect.
>
> Thanks for the observation. We agree that the benefits of CARS are most pronounced when an efficient incremental prefix-validity oracle is available. However, CARS does not require a perfect oracle: it can operate with any oracle that soundly discards a subset of invalid prefixes, even if it admits some invalid continuations.
>
> In particular, the oracle may conservatively allow tokens whose invalidity is only detected at a later stage; correctness (i.e., exact sampling) is preserved regardless. This observation induces a natural spectrum between ARS and CARS. At one extreme, when no prefixes can be ruled out until full sequences are evaluated (e.g., for global or black-box constraints), CARS becomes equivalent to ARS. At the other extreme, when constraints are fully prefix-checkable (e.g., CFGs), CARS achieves maximal pruning.
> Some settings lie between these extremes, where partial prefix information is available. In these cases, CARS degrades gracefully: weaker oracles reduce efficiency gains but do not affect correctness or applicability.
>
> ---
>
> > **C2.** I also think some of the empirical claims ... makes certain comparisons less complete than ideal.
>
> We agree that directly measuring distance to the true constrained distribution would be ideal. In practice, however, this is intractable: the constrained distribution is defined implicitly over a combinatorial (often unbounded) sequence space, and its normalization constant cannot be computed or queried exactly. As a result, exact evaluation of distances such as KL divergence to the true distribution is not feasible.
> Following prior work [1, 2], we instead estimate empirical KL divergence from samples as a proxy. While indirect, this metric is standard in the literature and provides a meaningful measure of how closely a method matches the target distribution in practice.
> Regarding the infeasibility of the AWRS-SMC baseline on our fuzzing benchmark, we acknowledge this leaves some comparisons incomplete and we can run these experiments on larger GPUs in the revision.
>
> [1] Park, K. et al. Grammar-Aligned Decoding. NeurIPS 2024.
>
> [2] Gonzalez, E.A. et al. Constrained Sampling for Language Models Should Be Easy: An MCMC Perspective. NeurIPS 2025.
>
> ---
>
> > **Q1.** I would appreciate a more direct discussion of how CARS compares to other possible exact approaches that exploit structured constraints ... and might admit dynamic-programming-style treatment.
>
> We thank the reviewer for this question. We interpret “highly regular” constraints as those expressible by finite automata or regular languages (we’ll gladly revise if our interpretation is incorrect).
>
> While such a structure can simplify constraint representation, it does not fundamentally change the complexity of exact sampling with LLMs. In particular, dynamic programming would require access to exact probabilities over future continuations in order to aggregate mass across equivalent states. However, autoregressive language models only provide conditional next-token distributions, and do not expose tractable marginals over full sequences or suffixes (w.r.t. the constraint). As a result, even when the constraint language is regular, exact sampling still requires reasoning over prefixes at the token level.
>
> ---
>
> > **C3.** In addition, although the trie overhead itself is small ... CARS can still fail badly there.
>
> > **Q2.** Also, could you discuss the failure mode ... discovered too late to help much?
>
> The primary reason acceptance can remain near zero is that, although most of the space is invalid, it is composed of many independent low-probability invalid regions that do not share a prefix. To illustrate, consider a constraint requiring the 20th token to be "the" with the prompt "generate a random sentence." This constraint can only be violated after 19 tokens have already been generated, and most sequences are likely to violate it if the LM has not been aligned with this constraint. Crucially, each rejected sequence removes only a tiny amount of probability mass, since the invalid regions branch out independently rather than sharing prunable prefixes – leading to very slow convergence. This is closely related to LM mismatch: when the model assigns negligible probability to the valid region, even aggressive pruning cannot compensate. We observe exactly this pattern in our experiments (Appendix G.5), where constraints requiring specific instructions mid-sequence cause all exact methods, including CARS, to struggle. In such settings, approximate methods may be the more practical choice.
>
> In terms of practical tradeoffs (see Lines 233–240), Theorem 3.3 shows that if CARS makes little progress over multiple iterations, one can disable further trie updates, effectively removing the additional trie-management overhead without significantly affecting future rejection rates.

---

> > ### Author Rebuttal · Reviewer_pFvs · 2026-04-05
> >
> > The authors answered my questions, I keep the same score.

---

### Official Review · Reviewer_Cjx5 · 2026-03-13

**Soundness:** 4
**Presentation:** 4
**Significance:** 3
**Originality:** 3
**Overall Recommendation:** 6
**Confidence:** 3

**Summary:**

This paper introduces CARS, which makes the rejection sampling much more efficient. The key idea: keep a trie of invalid prefixes you've already seen, then use that to reweight future samples. Instead of repeatedly hitting the same dead ends, the method learns what not to try. The result is higher acceptance rates over time.

Tests on grammar-based fuzzing, molecular generation, and text-to-SQL show it beats standard rejection sampling and adaptive variants on efficiency. Compared to approximate methods like GCD, MCMC, or AWRS, it also gets closer to the true distribution—or just performs better downstream.

**Compliance With Llm Reviewing Policy:**

Affirmed.

**Final Justification:**

I maintain my score.

This paper introduces CARS, a conceptually simple yet highly effective method for improving the efficiency of rejection sampling by caching and reweighting invalid prefixes using a trie structure. The key insight—learning from past rejected samples to avoid repeatedly exploring the same infeasible regions—is elegant and practically impactful. The approach preserves correctness while substantially improving acceptance rates over time.

The empirical evaluation is strong and diverse, spanning grammar-based fuzzing, molecular generation, and text-to-SQL. Across settings, CARS consistently improves efficiency over standard and adaptive rejection sampling, and compares favorably to approximate alternatives such as GCD, MCMC, and AWRS, often achieving better fidelity to the true distribution or stronger downstream performance.

My clarification question during review was addressed in the rebuttal, and the response reinforced my confidence in the method’s correctness and positioning. Overall, I find the paper technically sound, original in its formulation, and significant in practice.

**Key Questions For Authors:**

What is the asymptotic time complexity per generated token in terms of the structural ambiguity (e.g., number of admissible tokens) of the grammar, and how does this compare to lower bounds for grammar-masking engines?

**Limitations:**

No, the discussion is in the weaknesses and the questions parts.

**Strengths And Weaknesses:**

### Strengths

- Improves on ARS in a smart way: instead of just remembering which whole samples failed, it tracks every invalid prefix it encounters and avoids all possible completions of those paths. Simple in hindsight, but actually pulling it off with prefix-checkable constraints takes some work.

- Tests on three very different tasks—fuzzing, constrained SMILES generation, and text-to-SQL—each with its own constraint structure and what "good diversity" means.

Runs experiments across multiple models (Llama-3.1-8B-Instruct, Qwen2.5-7B/14B) and pits CARS against a mix of exact and approximate methods: RS, ARS, RSFT, GCD, MCMC, AWRS.

### Weaknesses
- Skips a natural baseline for text-to-SQL: PICARD and other constrained decoders. Even if they're approximate, seeing how they stack up on execution accuracy would've been useful context.

---

> ### Author Rebuttal · Authors · 2026-03-31
>
> > **C1.**  Skips a natural baseline for text-to-SQL: PICARD and other constrained decoders. Even if they're approximate, seeing how they stack up on execution accuracy would've been useful context.
>
> We thank the reviewer for highlighting this. PICARD is indeed a natural point of comparison for constrained text-to-SQL generation, and we investigated integrating it as a baseline. However, we encountered several challenges that make a direct comparison difficult in our setting.
>
> First, PICARD performs top-k (k=2) constrained beam search using a domain-specific incremental SQL parser that supports schema-aware semantic checks (e.g., column-table membership, alias validation) beyond what a context-free grammar can express, making it a fundamentally different constraint enforcement mechanism. Second, PICARD was designed for and evaluated with fine-tuned T5 models that produce SQL in a specific serialization format; community efforts to adapt it to decoder-only architectures have required fine-tuning to match this expected format [1, 2], and since our evaluation uses zero-shot prompting with decoder-only LLMs, a fair integration would require either fine-tuning our models or substantially re-engineering the parser. Third, the PICARD codebase presents practical compatibility barriers (a Python 3.7 environment, Haskell-based parser with a Thrift communication layer tightly coupled to T5's generation pipeline).
>
> That said, we note that our baselines already include GCD at temperature 1.0, which applies the same token-masking principle as PICARD but over the full vocabulary rather than only top-2 tokens, as well as state-of-the-art approximate sampling methods [3, 4].
>
> We do agree that evaluating constrained beam search as an additional baseline could provide useful context, and we can explore this in the revision.
>
> [1] https://github.com/ServiceNow/picard/issues/124
>
> [2] https://github.com/ServiceNow/picard/issues/5
>
> [3] Lipkin, B. et al. Fast Controlled Generation from Language Models with Adaptive Weighted Rejection Sampling. COLM 2025.
>
> [4] Gonzalez, E.A. et al. Constrained Sampling for Language Models Should Be Easy: An MCMC Perspective. NeurIPS 2025.
>
> ---
>
> > **Q1.** What is the asymptotic time complexity per generated token in terms of the structural ambiguity (e.g., number of admissible tokens) of the grammar, and how does this compare to lower bounds for grammar-masking engines?
>
> We are not sure which specific lower bounds the reviewer is referring to, but  we can describe CARS's complexity in more detail. CARS builds on top of existing GCD (grammar constrained decoding) implementations and inherits their complexity for grammar processing. In theory, GCD’s per-token cost is proportional to the full vocabulary size, since the LM itself returns a score vector over all tokens that must be appropriately masked. However, in practice the expensive operations are limited: only lightweight steps (masking, converting logits to probabilities) iterate over the full vocabulary. The actual parsing – which is more time-consuming – is in theory proportional to the number of admissible tokens, but GCD engines use clever pre-processing approaches to reduce this overhead and process multiple “equivalent” (wrt to parsing) tokens efficiently. Now to the complexity added by CARS. CARS itself adds negligible overhead on top of GCD: trie operations (node creation, probability propagation) are performed only for actually sampled tokens, therefore accounting for just 0.3% of total runtime as shown in Appendix F (i.e., the asymptotic complexity is effectively unchanged). We note that we did not implement parsing ourselves but rely on the LLGuidance library [1], whose performance characteristics are documented in [2]. While we chose this particular library, CARS is agnostic to the GCD backend – our implementation also supports GreatGramma [3], and any other grammar-masking engine can be incorporated straightforwardly.
>
> [1] https://github.com/guidance-ai/llguidance
>
> [2] https://guidance-ai.github.io/llguidance/llg-go-brrr
>
> [3] Park, K. et al. Flexible and Efficient Grammar-Constrained Decoding. ICML 2025.

---

> > ### Author Rebuttal · Reviewer_Cjx5 · 2026-04-02
> >
> > My concerns are addressed and thanks the detailed responses.

---

### Decision · Program_Chairs · 2026-04-30

**Decision:**

Accept (regular)

**Comment:**

This paper proposes CARS, an exact constrained sampling method that improves the efficiency of rejection sampling by caching invalid prefixes in a trie and reweighting future draws to avoid revisiting known-invalid regions. Reviewers were broadly enthusiastic about the paper’s core idea, theoretical guarantees, and empirical results. In particular, they highlighted the combination of exactness, monotonic acceptance-rate improvement, and strong performance across diverse domains, including fuzzing, molecular generation, and text-to-SQL. The work was viewed as technically strong, well executed, and practically relevant. Overall, I find this to be a strong paper. It makes a clean algorithmic contribution with solid theory, convincing experiments, and clear practical value. I recommend acceptance.